# Unrestrained growth of correctly oriented microtubules instructs axonal microtubule orientation

**Maximilian AH Jakobs[1,2]\*, Assaf Zemel[3], Kristian Franze[1,4,5]\***

[1]Department of Physiology, Development and Neuroscience, University of Cambridge, Cambridge, United Kingdom; [2]DeepMirror, Cambridge, United Kingdom; [3]Institute of Biomedical and Oral Research, and the Fritz Haber Center for Molecular Dynamics, Hebrew University of Jerusalem, Jerusalem, Israel; [4]Institute for Medical Physics, Friedrich-Alexander-Universität Erlangen-Nürnberg, Erlangen, Germany; [5]Max-Planck-Zentrum für Physik und Medizin, Erlangen, Germany

**Abstract** In many eukaryotic cells, directed molecular transport occurs along microtubules. Within neuronal axons, transport over vast distances particularly relies on uniformly oriented microtubules, whose plus-ends point towards the distal axon tip (anterogradely polymerizing, or plus-end-out). However, axonal microtubules initially have mixed orientations, and how they orient during development is not yet fully understood. Using live imaging of primary *Drosophila melanogaster* neurons, we found that, in the distal part of the axon, catastrophe rates of plus-end-out microtubules were significantly reduced compared to those of minus-end-out microtubules. Physical modelling revealed that plus-end-out microtubules should therefore exhibit persistent long-term growth, while growth of minus-end-out microtubules should be limited, leading to a bias in overall axonal microtubule orientation. Using chemical and physical perturbations of microtubule growth and genetic perturbations of the anti-catastrophe factor p150, which was enriched in the distal axon tip, we confirmed that the enhanced growth of plus-end-out microtubules is critical for achieving uniform microtubule orientation. Computer simulations of axon development integrating the enhanced plus-end-out microtubule growth identified here with previously suggested mechanisms, that is, dynein-based microtubule sliding and augmin-mediated templating, correctly predicted the long-term evolution of axonal microtubule orientation as found in our experiments. Our study thus leads to a holistic explanation of how axonal microtubules orient uniformly, a prerequisite for efficient long-range transport essential for neuronal functioning.

**\*For correspondence:**
max@deepmirror.ai (MAHJ);
kf284@cam.ac.uk (KF)

## Editor's evaluation

How axons form and maintain uniformly plus-end-out microtubules is an essential question in neuronal cell biology. Franze and colleagues used solid imaging and modeling approaches to provide important insights into the mechanisms controlling microtubule polarity in cultured *Drosophila* axons. They conclude that reduced catastrophe of the plus-end-out microtubules in the axon tip is critical for preferential plus-end-out microtubule growth and establishing the uniform microtubule polarity.

## Introduction

Symmetry breaking is critical for many biological systems. An organism starts off as a single round cell that divides and differentiates into many cells, tissues, and organ systems. The neuron, with its

**eLife digest** For humans to be able to wiggle their toes, messages need to travel from the brain to the foot, a distance well over a meter in many adults. This is made possible by neurons, the cells that form the nervous system, which transmit electrical signals along long extensions called 'axons'. Axons can only transmit signals if all the required molecules, which are produced in a part of the neuron known as the cell body, are ferried to the ends of the axons. This ferrying around of molecules is carried out by long, filamentous molecules called microtubules, which act as a directed carrier system, shuttling molecules along the axon, either towards or away from the cell body.

Microtubules can be thought of as asymmetrical rods. One end – known as the plus end – is dynamic and can undergo growth or shrinkage, while the other end – called the minus end – is stable. For transport along the axon to happen efficiently, microtubules in the neuron need to be oriented with their plus end pointing towards the ends of the axon. Microtubules in growing neurons develop this orientation, but how that is achieved is not fully understood.

To understand the basis of this cellular phenomenon, Jakobs, Zemel and Franze examined the behaviour of microtubules in developing neurons from fruit fly larvae. A fluorescent protein, which emits light when the microtubules are growing, helped the researchers visualise the plus end of microtubules, the microtubule orientation, and their growth in developing axons. This experiment showed that microtubules that had their plus end pointing towards the axon end shrank more slowly than those with the opposite orientation, leading them to grow longer. This resulted in a higher proportion of the correctly-oriented microtubules in the axon.

Treating the neurons with Nocodazole, a chemical that disrupts microtubule growth, or with sodium chloride, which changes the osmotic pressure, caused the microtubules that were oriented with their plus end towards the axon to grow less, and disrupted the uniform orientation of the microtubules in the axon.

The next step was to determine whether specific axonal proteins such as p150 – a protein that is enriched at the tip of the axon and decreases microtubule shrinkage rates – are involved in this process. Reducing the levels of p150 in fruit flies using molecular and genetic methods resulted in microtubules with their plus end pointing towards the axon tip shrinking faster, reducing the proportion of microtubules with this orientation in the axon. This role of proteins enriched in the axonal tip, along with previously discovered mechanisms, explains how microtubules align unidirectionally in axons.

These findings open new avenues of research into neurodegenerative diseases like Alzheimer's and Parkinson's, which might manifest due to a breakdown of transport along microtubules in neurons.

branched dendrites and sometimes exceedingly long axon, is one of the least symmetric cells found in animals. Axons connect neurons with distant targets and thus enable long-distance signal transmission throughout the body at high speed.

The enormous length of axons, which can extend over several meters in some vertebrate species, poses substantial logistical challenges. RNA, proteins, and organelles originating in the cell body need to be actively transported down the axon. Transport occurs along microtubules (MTs), which are long, polarized polymers that undergo stochastic cycles of growth and shrinkage (*Figure 1A*). Motor proteins transport cargo either towards a MT's dynamic (i.e., growing and shrinking) plus-end or the more stable minus-end (*Jiang et al., 2018*; *Jiang et al., 2014*).

In immature axons, MT orientation is mixed, with 50–80% of all MTs pointing with their plus-end-out (*del Castillo et al., 2015*; *Yau et al., 2016*). During early neuronal development, the fraction of plus-end-out axonal MTs increases (*del Castillo et al., 2015*; *Yau et al., 2016*). In mature axons, ~95% of all MTs point in the same direction (plus-end-out) (*Baas et al., 1989*; *Heidemann et al., 1981*), enabling polarized transport (*Millecamps and Julien, 2013*). Deficits in polarized transport have been associated with human neurodegenerative diseases, such as Alzheimer's and Parkinson's diseases (*Millecamps and Julien, 2013*). Despite the importance of polarized transport in neuronal axons, the mechanisms that establish and maintain MT orientation are still not fully understood (*Baas and Lin, 2011*; *Conde and Cáceres, 2009*; *Kapitein and Hoogenraad, 2011*).

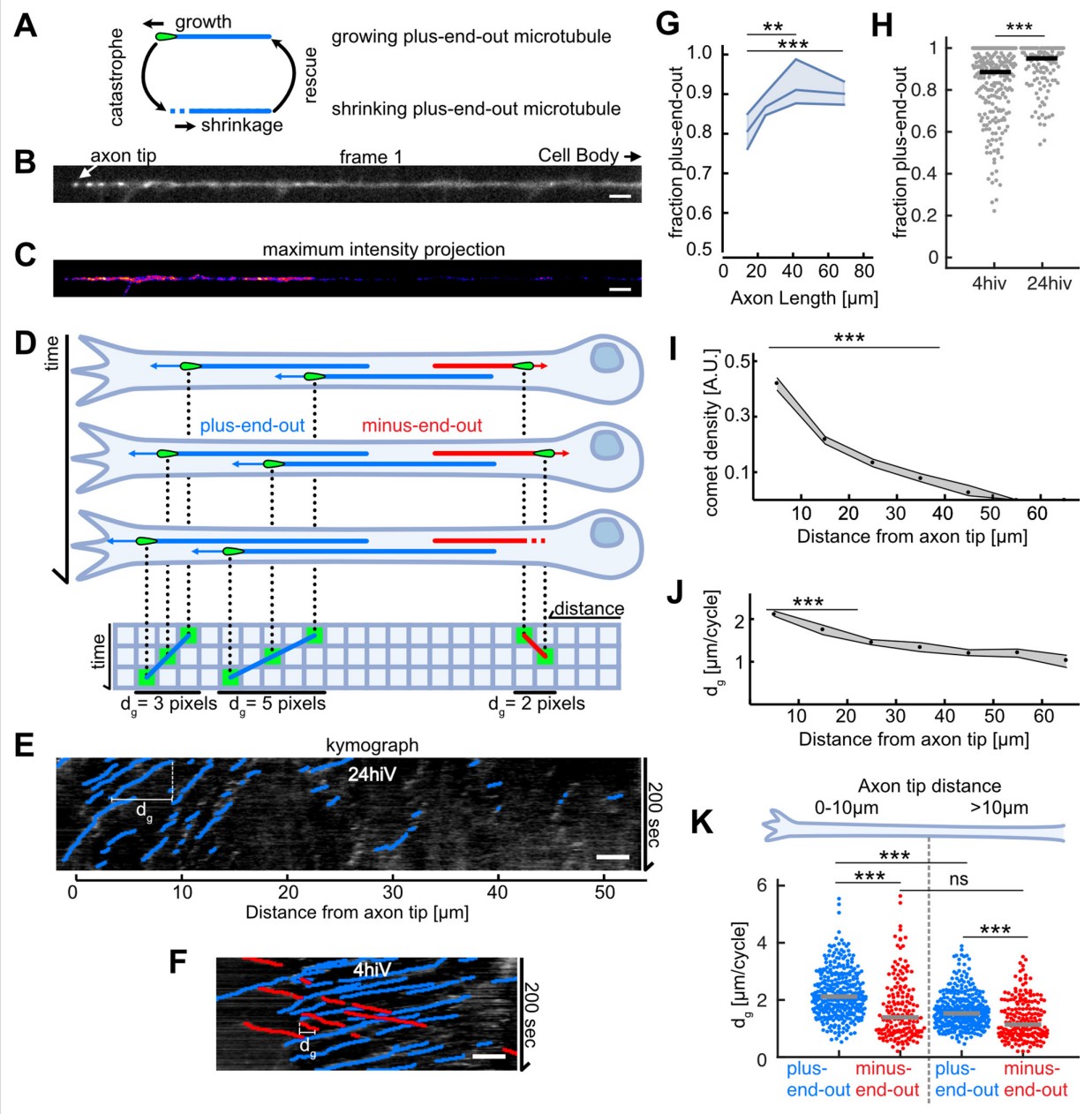

**Figure 1.** Axonal microtubule (MT) orientation increases over time and MT growth is enhanced at axon tips. (**A**) Schematic depicting the MT growth and shrinkage cycle. MTs grow until they undergo a catastrophe, which initiates MT shrinkage, and they start growing again after a rescue event. During growth (but not during shrinkage), EB1 localizes to MT tips. (**B**) First frame of a live cell imaging movie of axonal EB1-GFP dynamics. Bright dots represent individual EB1-GFP puncta, which label growing MT plus-ends. (**C**) Maximum intensity projection of a 200-s-long movie depicting EB1-GFP dynamics in a *Drosophila melanogaster* axon. EB1-GFP density is increased towards the tip. (**D**) Schematic showing how EB1-GFP live imaging movies were visualized and analysed using kymographs. The growing tips of plus-end-out MTs (blue) and minus-end-out MTs (red) were fluorescently labelled with EB1-GFP (green tear drop shaped 'comets'). The same axon is shown at three different time points; MTs grow at their plus-end, where EB1-GFP is located. The axonal intensity profiles of all time points are plotted underneath each other, resulting in a space-time grid called 'kymograph'. Connecting puncta between consecutive kymograph lines with blue/red lines yields the overall displacement $d_g$ for individual MT growth events. Note that the red minus-end-out MT stops growing in the second frame and shrinks in the third frame (**E**) Kymograph of an axon 24 hr post plating showing EB1-GFP dynamics analysed with *KymoButler* (*Jakobs et al., 2019*). Lines with a positive slope (blue, left to right upwards) are MTs growing with their plus-end towards the axon tip, lines with a negative slope (red, left to right downwards) are MTs growing away from the tip. Horizontal bars indicate the growth lengths ($d_g$) for individual MT growth cycles. (**F**) Kymographs of axonal processes expressing EB1-GFP analysed with *KymoButler* 4 hr post plating (**G**) MT orientation as a function of axon length. Longer axons exhibit a more pronounced plus-end-out MT orientation ($p<10^{-5}$, Kruskal-Wallis test, **$p<0.01$, ***$p<0.001$ for pairwise comparisons, Dunn-Sidak post hoc test). (**H**) MT orientation at 4 and 24 hiv (hours in vitro). MT orientation increased with time

*Figure 1 continued on next page*

*Figure 1 continued*

(p<10⁻⁴, Wilcoxon rank sum test). (**I**) EB1-GFP comet density as a function of the distance from the axon tip. Most MT polymerization occurred near the advancing axon tip. (N Axons = 353, 20 biological replicates from 20 different experiment days, p<10⁻²⁰, Kruskal-Wallis test, p<10⁻⁷ for pairwise comparisons between bins 1–2, 3, and 4, Dunn-Sidak post hoc test). Shown are median±95% confidence interval. (**J**) Added length per MT growth cycle $d_g$ as a function of distance from the axon tip. MTs grew longer in the vicinity of the axon tip (p<10⁻²⁰, Kruskal-Wallis test, p<10⁻⁷ for pairwise comparisons of either bin 1 or 2 with any other bin, Dunn-Sidak post hoc test). (**K**) $d_g$ for plus-end-out (blue) and minus-end-out (red) MTs grouped for growth in the distalmost 10 μm of the axon tip, and further away than 10 μm from the axon tip. Each dot represents the average of one axon in the respective region, grey lines indicate median values. With $d_g$ 2.11 [2.04, 2.16] μm/cycle (bootstrapped median [95% confidence interval]), plus-end-out MTs near the axon tip grew significantly longer than minus-end-out MTs ($d_g$ = 1.39 [1.27, 1.50] μm/cycle) and MTs located further away from the tip ($d_g$ = 1.53 [1.47, 1.59] μm/cycle, plus-end-out, $d_g$ = 1.16 [1.03, 1.33] μm/cycle, minus-end-out) (N=346 (plus-end-out close to tip), 343 (plus-end-out away from tip), 177 (minus-end-out close to tip), 194 (minus-end-out away from tip) axons), 20 biological replicates; p<10⁻³⁰, Kruskal-Wallis test followed by Dunn-Sidak post hoc test; ***p<10⁻⁴. Scale bars: 3 μm.

The online version of this article includes the following figure supplement(s) for figure 1:

**Figure supplement 1.** EB1 dynamics as a function of the distance from the axon tip.

MTs in post-mitotic neurons are not attached to the centrosome (*Kuijpers and Hoogenraad, 2011*). Nucleation of new MTs occurs from MT organizing centres (MTOCs) such as somatic Golgi (*Mukherjee et al., 2011*) through elongation of severed pieces (*Yu et al., 2008*) or de novo polymerization alongside existing MTs (*Nguyen et al., 2014*; *Sánchez-Huertas et al., 2016*). These newly formed MTs often orient in the same direction as existing ones, enforcing any pre-existing orientation bias (*Mattie et al., 2010*; *Mukherjee et al., 2020*). Pre-existing biases are furthermore enhanced by selective stabilization of MTs through TRIM46-mediated parallel bundling (*van Beuningen et al., 2015*). However, without pre-existing biases these mechanisms by themselves cannot explain the robust plus-end-out orientation of MTs in mature axons.

Furthermore, in axons, short MTs pointing with their minus-end away from the cell body can be transported towards the cell body (i.e., away from the tip) by cytoplasmic dynein (*del Castillo et al., 2015*; *Rao et al., 2017*), thus potentially clearing the axon of minus-end-out MTs. To test whether this mechanism is sufficient to establish uniform MT orientation in axons, we previously designed computer simulations of dynein-mediated MT sliding in neurons. However, while MTs in the distal axon were oriented mostly with their plus-end away from the cell body (*Jakobs et al., 2020*; *Jakobs et al., 2015*), our simulations failed to explain the longer-term plus-end-out orientation of MTs in the proximal axon and the gradual establishment of a uniform plus-end-out MT orientation throughout the axon seen in experiments (*Yau et al., 2016*). Thus, our simulations suggested that additional mechanisms are needed to establish the uniform plus-end-out orientation of MTs in neuronal axons.

Here, we investigated MT growth behaviours along *D. melanogaster* axons and discovered increased plus-end polymerization of plus-end-out MTs near the advancing axon tip, which depended on the presence of MT anti-catastrophe protein gradients in the distal axon. A stochastic model of MT dynamics suggested that this growth bias leads to unbounded growth of these MTs, while minus-end-out MTs exhibit bounded growth and remain short. Experiments and computer simulations confirmed that this selective MT growth bias is critical for uniform axonal MT orientation. Integrating previously identified mechanisms with the decreased plus-end-out MT catastrophe rate discovered here led to a model explaining how uniform MT orientation is achieved in developing neuronal axons.

## Results

### Correctly oriented MTs add more length per growth cycle

To investigate how MT orientation in neuronal axons becomes biased, we first cultured acutely dissociated neurons from the *D. melanogaster* larval CNS (*Egger et al., 2013*) and quantified MT growth in axons. We used *Drosophila* lines expressing the fusion protein EB1-GFP, which labels growing MT plus-ends with bright 'comets' (*Figure 1A–C*; *Sánchez-Soriano et al., 2010*; *Stepanova et al., 2003*). The distance over which a comet moves in the axon is equal to the overall length $d_g$ that is added to an MT between the start of its growth cycle and the growth cycle's end (*Figure 1A*), at which usually a 'catastrophe' occurred, leading to MT shrinkage (*Figure 2*). The direction of growth reveals whether an MT is oriented with its plus-end away from (plus-end-out) or towards (minus-end-out) the cell

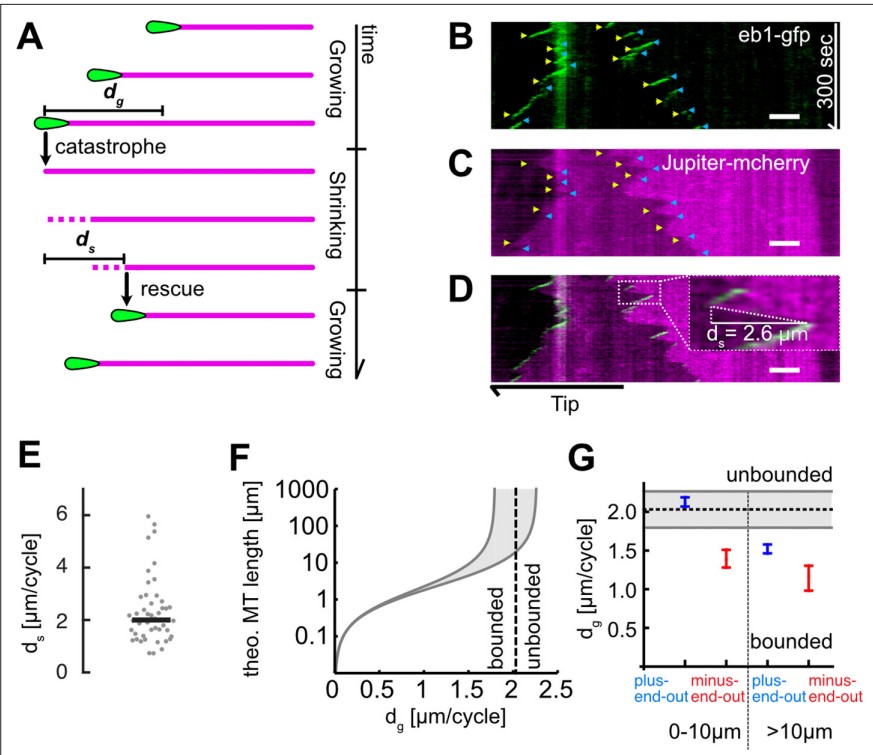

**Figure 2.** Microtubule (MT) length depends on added length per growth cycle. (**A**) Schematic highlighting the assumptions of our two-state master equation model. MTs were assumed to occupy either a growing or shrinking state. During a growth cycle, the average MT length increases by $d_g$, during a shrinkage cycle, the MT length decreases by $d_s$. Additionally, MTs were able to stochastically switch between the two states as shown in *Figure 1A*. (**B–D**) Kymographs from a *Drosophila melanogaster* axon that expressed (**B**) EB1-GFP (green) and (**C**) Jupiter-mCherry, a tubulin label (magenta). Individual MT shrinkage events, visible as (**C**) fluorescent edges and (**D**) dashed white lines in the kymograph, yielded MT shrinkage lengths per cycle $d_s$. Yellow and blue arrow heads in (**B**) and (**C**) indicate start and end points of an individual shrinkage event, respectively, and the inset in (**D**) highlights an individual shrinkage event. Scale bars: 3 µm. (**E**) Average $d_s$ values for $N$=47 axons (3 biological replicates; median: 2.03 [1.80, 2.26] µm) (bootstrapped median [95% confidence]). (**F**) Plot of the estimated overall MT length $l_{MT}$ as a function of $d_g$. The two solid black curves indicate the lower and upper bounds of the average MT lengths for a given $d_g$ with $d_s$ = 1.80 or 2.26 µm. One can separate two regimes, 'unbounded' and 'bounded' growth, separated by the dashed grey line. (**G**) Plot of $d_g$ as a function of MT orientation and localization showing median and 95% confidence intervals. Plus-end-out MTs close to the tip were considerably more likely to exhibit unbounded growth than plus-end-out MTs further away from the tip and minus-end-out MTs.

body. Time-lapse movies of EB1-GFP comets were converted into kymographs and analysed using *KymoButler* (*Jakobs et al., 2019*; *Figure 1D–F*).

The fraction of plus-end-out MTs increased over time and with increasing axonal length (*Figure 1G–H*), confirming that MT orientation increases during development (*del Castillo et al., 2015*; *Yau et al., 2016*). Most MT growth events (~66%) were found within the first 20 µm from the advancing axon tip (*Figure 1A and J*). MT growth lengths per cycle, $d_g$ were significantly higher near the axon tip compared to further away from it (*Figure 1K*). Furthermore, plus-end-out MTs added significantly more length per growth cycle than minus-end-out MTs, with the highest difference between plus-end-out and minus-end-out MTs (~0.5 µm/cycle) found within the first 10 µm from the axon tip ($d_g$ (plus-end-out)=2.11 [2.04, 2.16] µm/cycle and $d_g$ (minus-end-out)=1.39 [1.27, 1.50] µm, bootstrapped median [95% confidence interval]) (*Figure 1L*).

Increases in MT growth lengths during a polymerization cycle $d_g = v_g f_g$ could either arise from an increased polymerization velocity $v_g$ or a decreased catastrophe frequency $f_g$ (or both). In *Drosophila* neurons, MT growth velocities $v_g$ were similar in all MTs irrespective of their orientation and position within the axon (~5 µm/min; *Figure 1—figure supplement 1C*). Additionally, EB1-GFP, which affects MT polymerization velocities, was not enriched at the axon tip (*Figure 1—figure supplement*

*1E*), and EB1-GFP comet lengths, which have previously been linked to MT polymerization speeds (*Hahn et al., 2021*), did not show significant variations along the axon (*Figure 1—figure supplement 1F*). However, catastrophe rates $f_g = 1/t_g$ were significantly lower (and thus polymerization times $t_g$ significantly longer) in plus-end-out MTs near the axon tip ($f_{g+} \sim 4 * 10^{-2} s^{-1}$ vs. $f_{g-} \sim 6 * 10^{-2} s^{-1}$ ; *Figure 1—figure supplement 1B*), suggesting that these MTs grew longer because of a decrease in their catastrophe frequencies.

Overall, the orientation of MTs that added more length per growth cycle (i.e., plus-end-out MTs) becomes the dominant MT orientation in developing axons, indicating a possible link between increased plus-end-out MT growth and the fraction of plus-end-out MTs in axons.

## Enhanced growth of plus-end-out MTs leads to unbounded growth

On long time scales, differences in MT growth length per growth cycle accumulate and thus affect the average MT length $l_{MT}$. To estimate whether the rather small differences in $d_g$ of ~0.5 μm/cycle might lead to biologically meaningful differences in the average expected MT lengths between plus-end-out and minus-end-out oriented MTs, we used a two-state master equation model of MT growth and shrinkage (see *Dogterom and Leibler, 1993* and supplemental methods for details). The model distinguishes two regimes (*Figure 2A*)

$$l_{MT} = \begin{cases} d_g d_s / (d_s - d_g) & d_s > d_g & (\text{`bounded' growth}) \\ \infty & d_s \leq d_g & (\text{`unbounded' growth}) \end{cases} \tag{1}$$

where $d_s$ = lost length per shrinkage cycle.

When $d_s \leq d_g$, the average length added to the MT exceeds the average shrinkage length per cycle so that an MT will exhibit net growth and elongate if physically possible in its confined environment (called 'unbounded' growth, as the function describing MT growth in this regime goes towards infinity). For $d_s > d_g$, however, growth is 'bounded' (i.e., the end points of this function are finite), and average MT lengths follow an exponential distribution with a mean of $d_s d_g/(d_s - d_g)$. In practice, this means that MTs with $d_g \geq d_s$ would grow until encountering a physical barrier (e.g., the distal end of the axon tip), while MTs with $d_s > d_g$ remain finite (e.g., approximately 2 μm with $d_s$ = 2 μm and $d_g$ = 1.5 μm).

We determined the MT shrinkage per cycle $d_s$ by co-expressing a Jupiter-mCherry fusion protein (a tubulin marker) together with EB1-GFP in *D. melanogaster* axons of larval primary neurons. MTs stopped growing when the GFP signal disappeared from their plus-end, indicating a catastrophe or pause event. Subsequent MT shrinkage was visualized by simultaneously imaging tubulin (Jupiter-mCherry) and quantified by tracing tubulin edges resulting from the shrinkage in the dual colour kymographs (*Figure 2B–D*). Axonal MT shrinkage lengths were $d_s$ = 2.03 [1.80, 2.26] μm/cycle (bootstrapped median [95% confidence interval], *Figure 2E*).

With this value for $d_s$, our model predicted the divergence of $l_{MT}$ (i.e., the change from bounded/finite growth to unbounded/infinite growth) at $d_g$ = 2.03 μm. This value $d_g$ corresponded to the lower end of the measured 95% confidence interval for the median $d_g$ = [2.04, 2.16] μm/cycle of plus-end-out MTs near the axon tip, but it was well above the 95% confidence interval for the median $d_g$ = [1.39, 1.48] μm/cycle of all other MTs (*Figure 2F*). The measured values of $d_g$ and $d_s$ hence suggested that plus-end-out-oriented MTs exhibit mostly unbounded growth within 10 μm from the axon tip while minus-end-out MTs within that range and any MT further away from the tip do not (*Figure 2F*). The decreased catastrophe frequency of plus-end-out MTs near the axon tip (*Figure 1—figure supplement 1A-C*) implied a higher chance of survival for plus-end-out MTs while leaving minus-end-out MTs labile, thereby establishing a bias for plus-end-out MT orientation.

## Enhanced growth of plus-end-out MTs is required for uniform plus-end-out MT orientation

To test if enhanced MT growth is indeed involved in biasing MT orientation, we first chemically decreased MT growth using nocodazole, a drug that disrupts MT polymerization. Nocodazole treatment led to decreased MT growth velocities $v_g$ (*Figure 4—figure supplement 3*). Alternatively, we

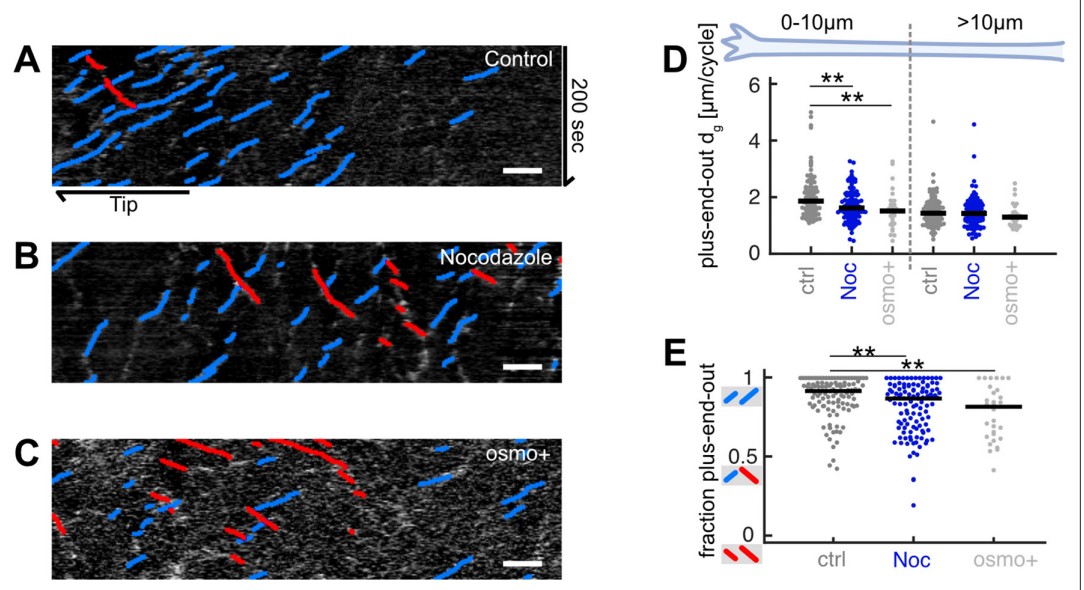

**Figure 3.** Decreasing microtubule (MT) growth leads to decreased axonal MT orientation. (**A–C**) Representative kymographs analysed with *KymoButler* (*Jakobs et al., 2019*) from axonal processes treated with (**A**) 0.025% DMSO (control) for 8 hr, (**B**) 5 μM nocodazole for 8 hr, (**C**) medium with increased osmolarity ('osmo+') for 22 hr. Growth of plus-end-out MTs is shown as blue lines, minus-end MTs are red. Scale bars = 3 μm. (**D**) Added MT lengths per growth cycle $d_g$ of plus-end-out MTs at the distalmost 10 μm from the axon tip and further away for control (N=107 axons from 5 biological replicates), nocodazole-treated axons (N=116 axons from 3 biological replicates), and axons cultured in osmo+medium (N=30, 2 biological replicates). At the axon tip, MT lengths increased significantly less per growth cycle in axons treated with nocodazole or osmo+media than controls (p<$10^{-4}$, Kruskal-Wallis test, **p<0.01 for pairwise comparisons, Dunn-Sidak post hoc test). (**E**) The fraction of plus-end-out MTs in the different groups. MT orientation was calculated by counting all MTs that grew away from the cell body (blue lines in kymographs) and dividing them by all growing MTs (blue and red) along the whole axons. This way, a kymograph with only blue lines gives a value of 1 while an equal number of blue and red lines yield a value of 0.5. MTs in axons treated with nocodazole or osmo+media were significantly less uniformly oriented than those in the control group, that is, they contained a larger fraction of MTs pointing with their plus-ends toward the cell body (red lines in (**A–C**)) (p<$10^{-4}$, Kruskal-Wallis test, **p<$10^{-2}$ for pairwise comparisons, Dunn-Sidak post hoc test).

also physically decreased MT growth by increasing the osmolarity of the cell culture medium through addition of NaCl (*Bray et al., 1991*; *Molines et al., 2020*). Here, MT catastrophe rates $f_g$ were significantly increased (*Figure 4—figure supplement 3*).

Both treatments led to significantly decreased plus-end-out MT growth $d_g = v_g/f_g$ at the axon tip (<10 μm), with $d_g < 1.71$ μm/cycle $< d_s$ (upper 95% confidence bound of median) (*Figure 3A–E*). Our model predicted that this change in $d_g$ should lead to a switch from previously unbounded to bounded growth of plus-end-out MTs near the axon tip (*Figure 3D*), thus reducing the bias towards plus-end-out MTs and decreasing MT orientation. In agreement, MTs within treated axons were overall significantly less uniformly oriented (*Figure 3D–E*), confirming an important role of the enhanced growth lengths per cycle of plus-end-out MTs in establishing axonal MT orientation.

## p150 protein gradient in axon tips promotes plus-end-out MT stabilization

However, why do plus-end-out MTs grow longer in the vicinity of the axon tip? Local gradients of MT growth-promoting factors could lead to an increase in plus-end MT growth in that region. Axon tips contain a multitude of different proteins and are highly compartmentalized (*Lowery and Van Vactor, 2009*). Locally enriched MT growth-promoting factors (of which there are many in the axon tip *Voelzmann et al., 2016*) include proteins stabilizing MTs, such as p150 by decreasing MT catastrophe rates (*Lazarus et al., 2013*; *Moughamian and Holzbaur, 2012*), CRMP-2 via promoting MT polymerization (*Fukata et al., 2002*; *Inagaki et al., 2001*), and TRIM46 by cross-linking MTs (*Rao et al., 2017*;

*van Beuningen et al., 2015*), as well as free tubulin required for MT polymerization and others (*Eng et al., 1999*).

The MT stabilizing protein p150, for example, affects MT catastrophe rates $f_g$ and nucleation rates but not growth velocities $v_g$ (*Lazarus et al., 2013*). As we observed decreased MT catastrophe rates but constant growth velocities of plus-end-out MTs towards axon tips (*Figure 1—figure supplement 1B,C*), we hypothesized that p150 is one of the key proteins involved. *Drosophila* has a p150 homologue which, similar as in murine neurons (*Moughamian and Holzbaur, 2012*), we found to be enriched in axonal but not in dendritic tips (i.e., tips of immature 'dendritic' processes), whose MT orientation is mixed and thus resembles that of vertebrate and immature *Drosophila* dendrites (*Hill et al., 2012*; *Figure 4A and C* and *Figure 4—figure supplement 1*). Accordingly, plus-end-out MT growth lengths per cycle $d_g$ and catastrophe rates $f_g$ were significantly higher in axons than in dendritic processes (*Figure 4—figure supplement 2E,G*), where $d_g$ = 1.42 [1.37, 1.39] µm/cycle remained smaller than $d_s$ = 2.03 [1.80, 2.26] µm/cycle. In agreement with our model, which predicted unbounded growth of plus-end-out MTs in axons but bounded growth in dendritic processes (*Figure 2*), dendritic processes exhibited mixed (50% plus-end-out) MT orientations (*Figure 4—figure supplement 2D*, *Hill et al., 2012*), corroborating a link between MT stabilizing protein gradients in the tip of axons, reduced MT catastrophe rates, unbounded growth, and overall MT orientation.

We next tested whether the observed p150 gradient is strong enough to lead to different MT growth lengths (via modulation of catastrophe rates, $d_g \sim 1/f_g$) for plus-end-out and minus-end-out MTs within 10 µm from the axon tip, the region where we observed the largest differences in $f_g$ (*Figure 1—figure supplement 1C*). We assumed that MT growth lengths per cycle at a distance $x$ from the axon tip can be described by a power law function $d_g(x) = A*p150(x)^\alpha$ of the p150 fluorescence intensity profile. A simultaneous fit to both plus-end-out and minus-end-out MTs demonstrated that the observed gradient is, in theory, indeed strong enough to cause different growth behaviours for plus-end-out and minus-end-out MTs in axonal tips (*Figure 4—figure supplement 4*).

To test this prediction further, we assessed MT dynamics in *wild-type*, *p150-RNAi* expressing larval neurons, and in heterozygous mutant larval neurons from *p150[1]/+* flies (*Figure 4*). *p150[1]* (also known as *Gl[1]*) mutants express a truncated p150-RNA transcript, which results in a dominant negative phenotype (*Plough and Ives, 1935*). Expressing *p150-RNAi* led to decreased p150 protein in axons (*Figure 4A–C*). Both the expression of *p150-RNAi* and of dominant negative *p150[1]/+* led to a significant increase in plus-end-out MT catastrophe rates (*Figure 4—figure supplement 3*) and thereby to a decrease (14–20%) in plus-end-out MT growth within 10 µm from the axon tip (*Figure 4G*). In agreement with our model, the overall axonal MT orientation was significantly decreased in both *p150-RNAi* and *p150[1]/+* axons compared to controls (*Figure 4H*). p150 is known to interact with dynein (*Karki and Holzbaur, 1995*). Expressing an RNAi against dynein heavy chain also led to decreased plus-end-out MT growth at the axon tip (25%) and decreased MT organization, resembling the results of p150 removal (*Figure 4—figure supplement 3*). Together, these data indicated that MT stabilizing or growth-promoting protein gradients at the axon tip do indeed have an important role in regulating the overall orientation of the axonal MT network.

## Kinesin 1 is required to establish p150 gradient

Previous work showed that p150 accumulation at axon tips depends on the activity of the MT-specific molecular motor protein kinesin 1 (*Moughamian and Holzbaur, 2012*; *Twelvetrees et al., 2016*), which preferentially enters axons over dendrites (*Tas et al., 2017*). Accordingly, disruption of kinesin 1 function with an RNAi treatment led to a 25% decrease in p150 fluorescence within 10 µm from the axon tip (*Figure 4—figure supplement 6A-D*). Again, the absence of a p150 gradient in these neurons led to a significant increase in MT catastrophe rates and decreased plus-end-out MT growth near the axon tip, and hence to an overall decrease in axonal MT orientation (*Figure 4—figure supplement 6E-I*), suggesting that gradients of MT stabilizing proteins at the tips of developing axons are critical for biasing axonal MT orientations.

## Uniform axonal MT orientation is established through a combination of MT sliding, templating, and unbounded growth

Our experiments suggested that a gradient of an MT growth-promoting factor localized at the axon tip is required for establishing uniform plus-end-out MT orientation in axons. To investigate the

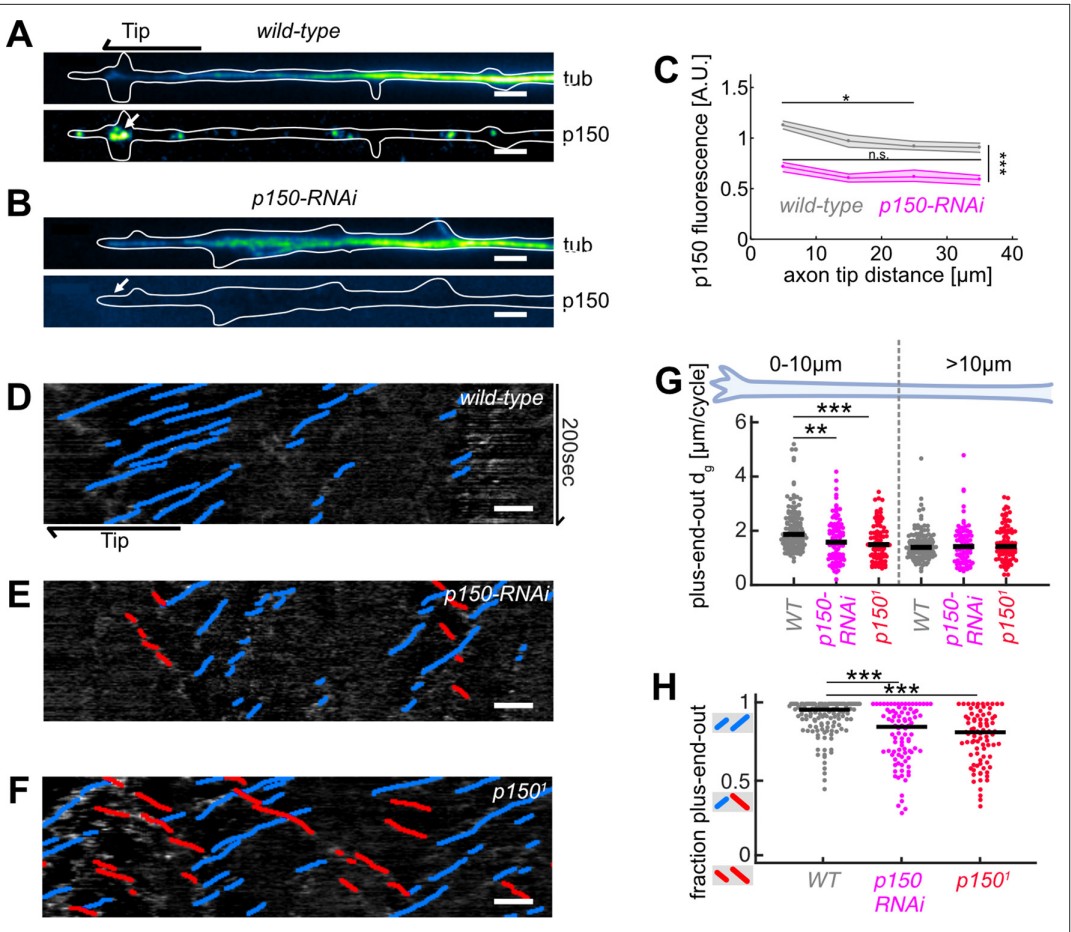

**Figure 4.** Abrogation of p150 function decreases microtubule (MT) growth and axonal MT orientation. (**A–B**) Tubulin (top) and normalized p150 (bottom) immunostaining of cultured *Drosophila melanogaster* larvae axonal processes of (**A**) controls and (**B**) neurons expressing *elav-gal4* UAS-driven *p150-RNAi*. Large p150 puncta were found clustered around the axon tip (arrow) in controls (**A**) but not in *p150-RNAi* axons (**B**). Scale bars = 2 µm (**C**) Normalized p150 fluorescence intensity as a function of distance from the axon tip for *wild-type* axons (*N*=83, 2 biological replicates) and *p150-RNAi* axons (*N*=111, 2 biological replicates). Lines represent median±95% confidence intervals for *wild-type* (grey) and *p150-RNAi* (magenta). p150 fluorescence intensities changed along the axon ($p<10^{-70}$; Kruskal-Wallis test). In *wild-type* axons, p150 was enriched at the axon tip (*$p<0.05$ between bin 1 and bin 3 or 4; pairwise comparisons with Dunn-Sidak post hoc test), but not in *p150-RNAi* expressing axons ($p>0.05$ for all pairwise comparisons). Overall, p150 expression levels were diminished in *p150-RNAi* axons compared to *wild-type* (***$p<10^{-7}$ for any pairwise comparison between conditions). (**D–F**) KymoButler output for kymographs of EB1-GFP expressed in (**D**) a *wild-type* axon, (**E**) an axon expressing p150-RNAi, and (**F**) an axon in a *p150$^1$/+* mutant background. Scale bars = 3 µm. Blue/red lines represent MTs with plus/minus-end-out orientation, respectively. (**G**) Plus-end-out MT added length per growth cycle $d_g$ for *wild-type* (*N*=85, 9 biological replicates), *p150-RNAi* (*N*=34, 3 biological replicates), and *p150$^1$/+* (*N*=83, 6 biological replicates). At the axon tip, MT growth lengths were significantly decreased in both *p150-RNAi* and *p150$^1$* conditions compared to controls ($p<10^{-9}$, Kruskal-Wallis test, ***$p<0.001$, *$p<0.05$, Dunn-Sidak post hoc test). (**H**) MT orientation along the whole axon for *wild-type*, *p150-RNAi*, and *p150$^1$/+*. MTs were less uniformly oriented in both *p150-RNAi* and *p150$^1$* axons ($p<10^{-9}$, Kruskal-Wallis test, ***$p<10^{-5}$ for pairwise comparisons with Dunn-Sidak post hoc test). Overall, axonal MT orientation was decreased after chemical, physical, and genetic perturbations of MT growth.

The online version of this article includes the following figure supplement(s) for figure 4:

**Figure supplement 1.** Normalized p150 immunostaining in axonal and dendritic processes.

**Figure supplement 2.** Microtubule (MT) growth and orientation is decreased in dendritic processes.

**Figure supplement 3.** Microtubule (MT) growth parameters for nocodazole and osmo+ treatments (**A, B**) and p150 knockdown (**C, D**).

*Figure 4 continued on next page*

*Figure 4 continued*

**Figure supplement 4.** The relationship between p150 protein concentration and the added length per microtubule (MT) growth cycle.

**Figure supplement 5.** Downregulation of dynein heavy chain expression decreases microtubule (MT) orientation and plus-end-out MT growth at axon tips.

**Figure supplement 6.** Disruption of kinesin 1 function reduces p150 concentration at axon tips and decreases microtubule (MT) orientation.

**Figure supplement 7.** Example calculation of a single axons' p150 profile.

significance of the locally biased MT growth in more detailed, we modelled the evolution of overall MT polarity along the axon using computer simulations. We extended our previously described simulations of MT-MT sliding, which led to sorting of MTs based on their orientation (*Jakobs et al., 2020*), to include two additional effects on axonal MTs: (1) enhanced stabilization of MTs at axon tips as found in this study and (2) a local biasing of MT nucleation (MT templating) that could originate from augmin (*Nguyen et al., 2014*; *Sánchez-Huertas et al., 2016*) or TRIM46 (*Rao et al., 2017*).

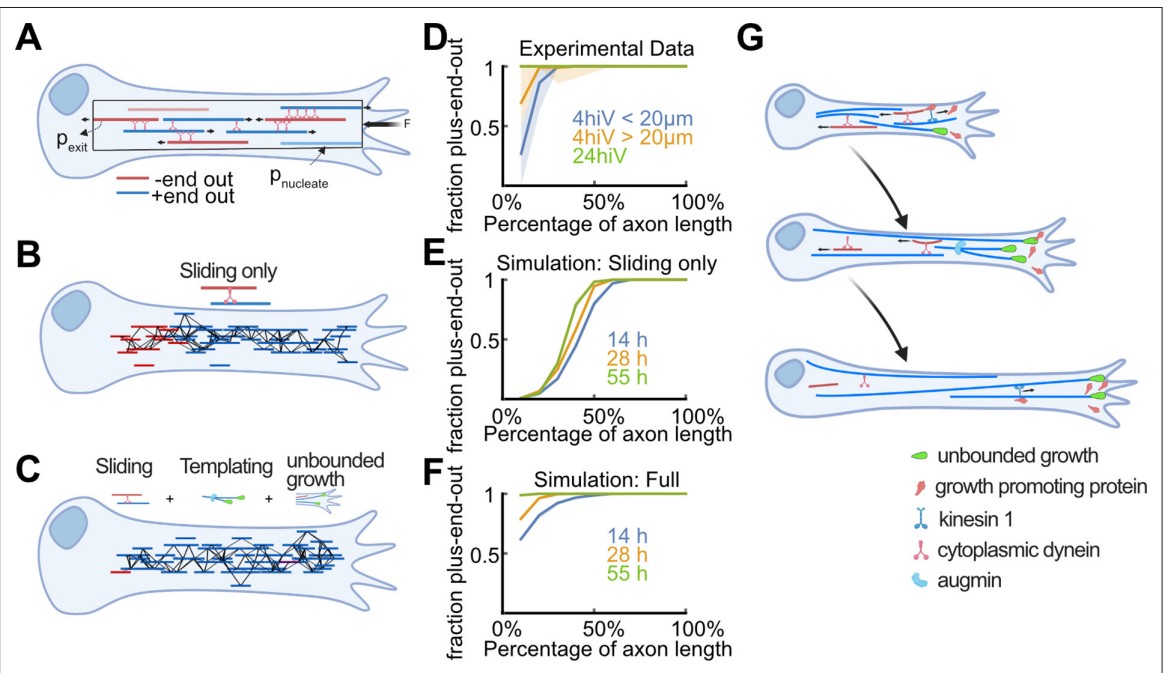

**Figure 5.** Biased stabilization of plus-end-out microtubules (MTs) is required to establish uniform axonal MT orientation. (**A**) Schematic showing the MT sliding simulation and its relevant parameters. (**B–C**) Simulation results of MT dynamics. (**B**) Snapshot of a simulated axon with dynein-based sliding of MTs minus-end-out MTs accumulated within the proximal axon. (**C**) Snapshot of an axon simulated with sliding, augmin templating (new MTs were likely oriented into the same direction as their surrounding ones), and biased nucleation of plus-end-out MTs at the tip. Much like in real axons, most MTs were oriented with their plus-end-out throughout the axon. (**D**) Experimental profiles of MT orientation along the (normalized) axon length for axons that were cultured 4 hr in vitro (4hiv), separated into less than 20 µm or more than 20 µm long, and 24 hr in culture (24hiv). Lines represent bootstrapped medians and 95% confidence intervals. (**E–F**) Simulation profiles of MT orientation along the axon. (**E**) Profiles obtained from simulations with dynein-mediated sliding only at three different simulation time points. MT orientation was graded along the axon but, unlike in the experimental profiles shown in (**D**), the proximal axon remained enriched with minus-end-out MTs. (**F**) MT orientation profiles obtained from simulations with sliding, templating, and biased nucleation of plus-end-out MTs at the axon tip. The observed gradual development of MT orientation along the axon is in excellent agreement with our experimental data (**D**). (**G**) Summary of proposed mechanism for establishing MT orientation in axons. Red and blue lines represent minus-end-out and plus-end-out MTs, respectively. Green drop shapes indicate unbounded MT growth into the axon tip for plus-end-out MTs. Kinesin 1 deposits MT growth-promoting proteins, such as p150, at axon tips (*Figure 4—figure supplement 4*), leading to local unbounded growth of plus-end-out MTs. Augmin templating and cell body-directed sliding of minus-end-out MTs further amplifies this bias. All three mechanisms together lead to a plus-end-out MT cytoskeleton.

The online version of this article includes the following figure supplement(s) for figure 5:

**Figure supplement 1.** Detailed results of computer simulations for different microtubule (MT) sorting models.

The simulation is detailed in the Materials and methods section and a schematic can be found in *Figure 5A*. Briefly, a cylindrical bundle of MTs is generated, and dynein motors are assumed to cross-link adjacent MTs. The orientation of each MT is randomly chosen. Force-velocity relationships are used to predict the exerted sliding velocities of the MTs, which are then used to calculate new MT positions iteratively. MTs were confined in the axon with a solid boundary at the axon tip and a semi-permeable boundary at the cell body. New MTs were added randomly along the bundle axis at a determined frequency. To mimic the effect of a concentration gradient of a catastrophe-inhibiting protein (such as p150) at the axon tip, we assumed that the frequency of newly added MTs is sampled from an exponential distribution that peaks at the axon tip. To account for augmin or TRIM46-induced local biases of MT nucleation, we assumed that the orientation of an added MT is dictated by the mean MT orientation in the region to which it is added.

Simulations incorporating dynein-induced MT sliding but lacking mechanisms (1) and (2) mentioned above resulted in steadily growing axons whose MT orientation profile was graded along their length, being enriched with plus-end-out MTs at the distal end and with minus-end-out MTs at the proximal (cell body) end (*Figure 5A, B and E* and *Figure 5—figure supplement 1*). The proximal domain of minus-end-out MTs grew in proportion to the axon length, as minus-end-out MTs from across the growing axon continually accumulated in that region despite their withdrawal into the cell body by dynein motors. Those simulations thus failed to reproduce the experimental observation of an enrichment of the proximal axon with plus-end-out MTs over time, which eventually leads to uniformly oriented axonal MTs (*Yau et al., 2016*; *Figure 5D*).

Similarly, the increased growth of plus-end-out MTs or templating alone, or any combination of two out of these three mechanisms, were also insufficient for establishing high fractions of plus-end-out MTs at the proximal axon as observed experimentally (*Figure 5—figure supplement 1*).

However, when we integrated sliding, templating, and biased MT growth at the axon tip in the simulation (assuming that the likelihood of exhibiting unbounded growth corresponds to a successful nucleation event), MTs gradually oriented uniformly plus-end-out across the entire length of the axon, recapitulating our experimental results (*Figure 5C, D and F* and *Figure 5—figure supplement 1*). Thus, our data suggest that multiple mechanisms – including biased growth of plus-end-out MTs near the axon tip identified in this study – need to work in unison to establish and maintain uniform axonal MT orientation.

## Discussion

We here found that MT growth is an important factor in the regulation of overall MT organization in the axon. Our experiments and modelling suggest that an enrichment of MT stabilizing/growth-promoting proteins at the advancing axon tip leads to a transition of MT growth from a bounded to an unbounded state for plus-end-out MTs. This growth transition is important for establishing uniform plus-end-out MT orientation from the cell body to the axon tip as found in mature axons. While previous studies suggested that MT dynamics are temporally and spatially constant during early axon formation (*Seetapun and Odde, 2010*), our results suggest that, at later stages of axon maturation, MT dynamics are heterogeneous (*Figure 1*).

Our chemical and physical perturbations of MT growth affected different aspects of MT dynamics. Nocodazole treatment decreased MT growth velocities while not affecting catastrophe frequencies; hyperosmotic solutions increased MT catastrophe frequencies but did not alter growth velocities (*Figure 4—figure supplement 3*). The effect of the hyperosmotic solution might potentially arise from a decrease in the available space in the distal axon for MTs to polymerize into (*Dogterom and Yurke, 1997*; *Franze, 2020*). However, NaCl can be toxic for neurons, so that the effect on MT growth could also be due to a general stress response (*Morland et al., 2016*). Either way, both treatments led to a decrease of the MT length added during each growth cycle, $d_g$, so that $d_g < d_s$, switching plus-end-out MT dynamics from unbounded to bounded growth, thus leading to a loss of uniform MT polarization along the axon (*Figure 3*).

The axon tip is highly enriched with growth-promoting factors such as p150 (*Lazarus et al., 2013*; *Moughamian and Holzbaur, 2012*), CRMP-2 (*Fukata et al., 2002*; *Inagaki et al., 2001*), TRIM46 (*Rao et al., 2017*; *van Beuningen et al., 2015*), and EB1 (*Ma et al., 2004*; *Morrison et al., 2002*). The anti-catastrophe protein p150, which we investigated here as an example, was concentrated at axon tips but was not enriched at dendritic tips (*Figure 4—figure supplement 1*). Enhanced stabilization

of plus-end-out MTs, leading to reduced catastrophe rates and thus unbounded growth essential for establishing uniform MT orientation, was only observed near axonal but not near dendritic tips (*Figure 4—figure supplement 2*), confirming that differences in the localization of MT growth-promoting proteins correlate with differences in MT growth. Perturbations of p150 led to increased catastrophe rates and decreased plus-end-out MT growth in the axon tip, and thus to decreased overall MT order in the axon (*Figure 3*). Interestingly, MT growth ($d_g$) scaled with the fourth power of the p150 concentration (*Figure 4—figure supplement 4*). This non-linear dependence could result from the requirement of several p150 proteins to form a complex to affect MT catastrophe rates, as previously hypothesized (*Lazarus et al., 2013*).

While p150 is mainly known for its role in the dynactin complex, which is an important cargo adapter protein complex for the molecular motor protein dynein (*Gill et al., 1991*), it also acts as a dynein-independent MT anti-catastrophe factor (*Lazarus et al., 2013*). Since p150 and dynein are functionally related, it is difficult to separate their individual contributions to stabilizing plus-end-out MT growth in the axon tip and cell body-directed sliding of minus-end-out MTs along the axon. However, it remains unclear whether p150 is required for dynein-mediated MT sliding (*Ahmad et al., 1998*; *Tan et al., 2018*; *Waterman-Storer et al., 1997*), and the *D. melanogaster* oocyte also contains an ordered MT cytoskeleton whose orientation is, presumably, maintained by p150 (*Nieuwburg et al., 2017*). Hence, while p150 is unlikely to induce unbounded MT growth alone, it emerges as a key contributor to the establishment of MT orientation.

In addition to its contribution to setting up the p150 gradient in axon tips (*Figure 4—figure supplement 6*)**,** kinesin 1 is also thought to slide MTs with their minus-end leading (*del Castillo et al., 2015*) and to be involved in dynein function (*Pilling et al., 2006*; *Rao et al., 2017*). Kinesin knockdown could thus not only prevent the accumulation of MT growth-promoting factors at the axon tip but also decrease kinesin 1-mediated minus-end-out MTs sliding into the distal axon (thereby increasing the fraction of plus-end-out MTs) and/or prevent retrograde dynein-mediated sliding of minus-end-out MTs (decreasing the fraction of plus-end-out MTs). We observed that disruption of kinesin 1 function led to a decrease in the fraction of plus-end-out MTs in the axon (*Figure 4—figure supplement 6*), suggesting that kinesin 1 does not contribute to the overall MT orientation via direct sliding but that it rather affects MT orientation mainly via localizing MT growth or nucleation-promoting proteins to axonal tips and/or via interactions with dynein-mediated sliding.

Finally, both kinesin 1 and p150/dynactin perturbations could potentially also affect MTOC localization in neurons. p150 was enriched at the tips of axonal but not of dendritic processes (*Figure 4—figure supplement 1*). In *Caenorhabditis elegans* neurons, MTOCs may be located to the tips of dendritic processes (*Liang et al., 2020*). Removal of dynactin, which initiates cell body-directed transport from axonal tips (*Moughamian and Holzbaur, 2012*), could lead to an increased number of MTOCs also at axon tips, thereby promoting growth and nucleation of MTs. However, our results showed a decrease in MT growth dynamics at axon tips after dynactin removal (*Figure 3*), indicating that the observed decrease in MT orientation was mostly due to decreased rather than promoted growth of axonal MTs in the axon tip.

Our simulations revealed that the uniform MT orientation along the axon cannot be understood solely based on dynein-mediated MT sliding: as axons extended, an increasing number of minus-end-out MTs from across the axonal shaft accumulated at the proximal axon. Additional mechanisms were needed to dilute the fraction of minus-end-out MTs. Amongst those, MT-templating (e.g., by augmin or TRIM46) and the biased MT growth mechanism identified in the present study both improved the MT orientation profile along the axonal shaft. Our experimental findings were best matched when all three mechanisms worked in concert (*Figure 5—figure supplement 1*).

We propose the following model explaining the spontaneous establishment of MT orientation in developing axons. Growth-promoting proteins accumulate at the axon tip due to MT plus-end-directed transport by kinesin motors (*Figure 4—figure supplement 6*). The resulting protein gradient leads to a local bias in MT growth (*Figure 1*, *Figure 3*), rendering plus-end-out MT growth into the axon tip unbounded (*Figure 2*). In theory, a finite pool of tubulin at axon tips would further contribute to this bias in plus-end-out MT growth by diminishing the amount of free tubulin available for MTs that polymerize towards the cell body. In contrast, short minus-end-out MTs are more prone to depolymerization and/or transport away from the tip by dynein-mediated cell body-directed sliding (*del Castillo et al., 2015*; *Rao et al., 2017*), thus contributing to the orientation bias of MTs in the

axon. The orientation bias is further enhanced by augmin-mediated templating or TRIM46-mediated parallel bundling of newly formed MTs to establish and maintain a fully organized MT cytoskeleton (see *Figure 5E* for a schematic summary). Together with cell process length-dependent MT accumulation (*Seetapun and Odde, 2010*), these mechanisms cooperate to build the uniformly oriented MT network that enables efficient long-range transport in neuronal axons. Future work will reveal whether other cellular systems use similar mechanisms to organize their cytoskeleton.

# Materials and methods

## Key resources table

| Reagent type (species) or resource | Designation | Source or reference | Identifiers | Additional information |
|---|---|---|---|---|
| Strain (*Drosophila melanogaster*) | EB1-GFP | *Bulgakova et al., 2012* | N/A | Flyline used to express EB1-GFP |
| Strain (*Drosophila melanogaster*) | Ubi EB1-GFP | *Shimada et al., 2006* | N/A | Complementary Flyline used to express EB1-GFP |
| Strain (*Drosophila melanogaster*) | Jupiter-mCherry | *Bergstralh et al., 2015* | N/A | Flyline used to label microtubules |
| Strain (*Drosophila melanogaster*) | P150[1] | Bloomington | RRID:BDSC_504 | p150 heterozygous mutant |
| Strain (*Drosophila melanogaster*) | P150-RNAi | Bloomington | RRID:BDSC_3785 | |
| Strain (*Drosophila melanogaster*) | khc-RNAi | Bloomington | RRID:BDSC_35770 | |
| Strain (*Drosophila melanogaster*) | Elav-gal4 | Bloomington | RRID:BDSC_458 | RNAi driver in neuronal cells |
| Antibody | Rb anti-p150 (rabbit polyclonal) | *Nieuwburg et al., 2017* | N/A | 1:500 |
| Antibody | Ms anti-alpha-tubulin (mouse monoclonal) | Abcam | Cat#: ab7291 RRID:AB_2241126 | 1:1000 |
| Recombinant DNA reagent | CellTracker | Invitrogen | Cat#: C2925 | ×1 |
| Antibody | CF633 anti-Rb (donkey anti-rabbit polyclonal) | Cambridge Bioscience | Cat#: BT20125 | 1:500 |
| Antibody | AF405 anti-Ms (donkey anti-mouse polyclonal) | Thermo Fisher Scientific | Cat#: ab175658 RRID:AB_2687445 | 1:500 |
| Software, algorithm | MATLAB | Mathworks | RRID:SCR_001622 | |
| Software, algorithm | Mathematica | Wolfram | RRID:SCR_014448 | |
| Software, algorithm | KymoButler | *Jakobs et al., 2019* | https://githlab.com/deepmirror/kymobutler; *deepMirror, 2019* | |
| Software, algorithm | ImageJ | *Schindelin et al., 2012* | RRID:SCR_003070 | |
| Software, algorithm | Neurite Tracer | *Pool et al., 2008* | RRID:SCR_016566 | |

## Fly stocks

MT plus-end dynamics were visualized with a transgenic fly line expressing EB1-GFP heterozygously under its endogenous promoter (*wh;+;eb1-gfp/tm6b*, gift from the Brown laboratory in Cambridge) (*Bulgakova et al., 2013*) or a fly expressing EB1-GFP under a *ubiquitin* promotor (*ubi:eb1-gfp;+;+*, gift from the St Johnston laboratory in Cambridge) (*Shimada et al., 2006*). Whole MTs were labelled with Jupiter-mCherry (*wh;if/cyo;Jupiter-mcherry*, gift from the St Johnston laboratory in Cambridge) (*Bergstralh et al., 2015*). Other stocks used: *p150[1]* (Bloomington # 504), *uas:p150-RNAi* (Vienna *Drosophila* Stock Center # 3785), *uas:khc-RNAi* (Bloomington # 35770), *khc27* (Bloomington # 67409), *khc17* (gift from the St Johnston laboratory in Cambridge). *uas* constructs were driven by *elav-gal4*

(Bloom# 458, *elav* is a neuron-specific promotor that ensures the construct is only expressed in the CNS; *Yannoni and White, 1997*) and transgenic lines were generated through standard balancer crossing procedures.

## Primary cell culture

Third instar larvae were picked 5–8 days post fertilization, and their CNS dissected similarly to *Egger et al., 2013*; *Sánchez-Soriano et al., 2010*. As described in *Egger et al., 2013*, the resulting primary culture comprised a mixture of terminally differentiated larvae neurons such as peripheral neurons alongside precursors cells and immature neurons of the adult fly brain. Thereby, the larval CNS lends itself to the study of a heterogenous population of neurons. The CNS tissue was homogenized and dissociated in 100 µl of Dispersion medium (Hank's Balanced Salt Solution [×1 HBSS, Life Technologies, 14170088] supplemented with Phenylthiourea (Sigma-Aldrich P7629, 0.05 mg/ml), Dispase (Roche 049404942078001, 4 mg/ml), and Collagenase (Worthington Biochem. LS004214, 1 mg/ml)) for 5 min at 37°C. The media was topped up with 200 µl of Cell Culture Medium (Schneider's Medium, Thermo Fisher Scientific 21720024) supplemented with insulin (2 µg/ml Sigma I0516) and fetal bovine serum (1:5 Thermo Fisher Scientific A3160801) and cells were spun down for 6 min at 650 rcf. The pellet was resuspended in Cell Culture Medium at 5 brains/120 µl. Cells were grown at 26°C for 1.5 hr in a droplet of 30 µl Cell Culture Medium in a glass bottom dish between a Concanavalin A-coated glass slide and an uncoated glass slide on top. Initially the cells were cultured with the coated coverslip facing down. After 1.5 hr the chambers were flipped so that cells that did not attach floated off to the opposite (uncoated) side. Culture times were: 4–26 hr (for measuring MT orientation profiles in short and long axons), 22–26 hr (for measuring MT dynamics, both Patronin-YFP and EB1-GFP), and 22–48 hr (for measuring MT dynamics in dendritic processes).

To measure the effect nocodazole has on MT orientation in axons, the medium was supplemented with 5 µM of nocodazole (dissolved in DMSO, Sigma-Aldrich M1404-2MG) approximately 12 hr post plating and 12 hr before measuring MT dynamics. The control cells were treated with 0.025% DMSO in culture medium. Treatment and corresponding controls were always run in parallel, and when possible, from the same fly stock. *Uas*-driven overexpression was controlled with a fly expressing both *elav::gal4* and *eb1-gfp* to control for the expression of gal4 protein.

To measure the effects of osmolarity changes in the surrounding medium, we increased the osmolarity of the culture medium by approximately 100 mOsm (from ~360 to ~460 mOsm, see also https://www.sigmaaldrich.com/GB/en/product/sigma/s9895) by adding 4 mg of NaCl to 1 ml culture medium. Cells were first cultured in normal media for 1.5 hr. Subsequently, the media was removed and replaced with either fresh media (control) or media supplemented with 4 g NaCl. Cells were again imaged after 22–26 hr post plating.

## Live imaging of MTs

All live imaging movies were acquired on a Leica DMi8 inverted microscope with a ×63 objective (oil immersion, NA = 1.4, Hamamatsu Orca Flash 2.0 camera) and at room temperature (22–25°C). To reduce autofluorescence during imaging, the culture medium was replaced with Live Imaging Solution (Thermo Fisher Scientific A14291DJ). Culture media was not replaced for imaging cells in nocodazole, DMSO, and osmo+ to enable measurement of MT dynamics in the chosen media. For EB1-GFP imaging, an image (exposure time 500 ms) was taken every 2 s for 70–150 frames depending on sample bleaching. When imaging both EB1-GFP and Jupiter-mCherry simultaneously, one image was taken every 3 s for 100 frames (exposure 500 ms). Lamp intensity was set to the lowest level that enabled visual identification of labels.

## p150 antibody staining

Twenty-four hr after plating, the cells were treated with 5 µM of CellTracker (Invitrogen C2925) dye for 30 min to label cells in green. Subsequently, cells were fixed in pre-warmed 4% paraformaldehyde (pH 7.2, 26°C) for 50 min. Post fixation, the cells were washed in PBS once and then incubated with mouse alpha-tubulin 1:1000 (Abcam ab7291) and rabbit glued/p150 antibody 1:500 (gift from the St Johnston laboratory, *Nieuwburg et al., 2017*) diluted in PBST (×2 phosphate-buffered saline (PBS, Oxoid BR0014G) tablets in 400 ml $H_2O$+1.2 ml Triton X-100)+0.01 g/ml bovine serum albumin at 4°C overnight (~14 hr). After two quick washes in PBS, the cells were incubated with the secondary

antibodies Alexa Fluor 647 (far-red, Thermo Fisher Scientific A-21236) and 405 (blue, Thermo Fisher Scientific A-31556) for 1.5 hr at room temperature. After another two quick PBS washes, the cells were mounted in Fluoromount (Thermo Fisher Scientific 00-4958-02) and imaged.

Images were analysed by drawing a line along axon processes from the base of the axon to its tip in the tubulin channel. The intensity profiles for all three channels (p150, tubulin, and CellTracker) were extracted and normalized by their respective median values, and the p150 channel was subsequently divided by the normalized CellTracker channel and the data binned in bins of 10 µm. Finally, the resulting binned *p150-RNAi* p150 profiles were normalized by the mean fluorescence of the respective *wild-type* control p150 fluorescence. The resulting profiles (per axon) were then pooled over all biological replicates and plotted in *Mathematica*. See *Figure 4—figure supplement 7* for a single cell workflow example.

## EB1-GFP dynamics

Kymographs of EB1-GFP tracks in *D. melanogaster* axons were generated by first using the Neurite Tracer plugin in ImageJ to draw lines along axons or dendrites from the centre of the cell body to the farthest EB1-GFP comet signal, that is, the distalmost growth event (*Pool et al., 2008*). Subsequently a custom *Mathematica* (https://wolfram.com) algorithm automatically generated kymographs from these lines by plotting the average pixel intensity of three adjacent pixels into rows of an image for each frame. The resulting image was then smoothed with a Gaussian kernel of size 3 and wavelet filtered to remove noise. Kymographs were analyzed with *KymoButler* and subsequently post-processed in *MATLAB* (https://mathworks.com). Tracks were removed in case: (1) they displaced less than two pixels along the x-axis, (2) they were slower than 1.5 µm/min, (3) they were faster than 20 µm/min, and (4) they were visible for less than four frames. Additionally, control experiments and their corresponding treatment condition were discarded if the control axons exhibited a mean orientation below 0.8 or average growth velocities below 2 µm/min. To account for outlier comets, the distance from the axon tip was calculated as the distance from the 0.95 quantile EB1-GFP comet.

MT minus-end polymerization is much slower than plus-end polymerization (*Strothman et al., 2019*). Since we observed similar growth velocities of cell body-directed and tip-directed MT growth, we are confident that we measured plus-end-out MT growth events in both directions rather than minus-end growth.

Note that, mature *D. melanogaster* dendrites in vivo exhibit a mixed MT orientation (*Stone et al., 2008*). However, we cultured neurons only up to 48 hr which might be too short to form fully developed dendrites and our minimal cell culture medium is likely lacking growth factors that would enable further differentiation to form fully minus-end-out dendrites. Additionally, vertebrate dendrites also appear to acquire their characteristic orientation over time (*Baas et al., 1989*).

## Jupiter-mCherry and EB1-GFP

Kymographs were prepared as for imaging EB1-GFP only (i.e., using Neurite Tracer). Individual shrinkage events were extracted by hand from the resulting kymographs using the ROI tool in *ImageJ* (https://imagej.net). The tracks were then analysed and plotted with *MATLAB* and *Mathematica*. Measuring MT shrinkage dynamics was only possible in regions of low tubulin content, for example, near the axon tip. We implicitly assumed that MT shrinkage depends neither on MT orientation nor on its position along the axon. However, experimental evidence suggests that a decrease in MT growth length correlates with an increase in shrinkage length (*Vasquez et al., 2017*), indicating that we likely overestimated MT lengths further away from the axon tip, therefore underestimating the difference between plus-end-out MTs at the tip and those further away from it.

## Statistics

For comparing two groups, the Wilcoxon rank sum test was used as implemented in MATLAB (https://www.mathworks.com/help/stats/ranksum.html). The standard error of the mean (s.e.m.) was calculated as $s.e.m. = \sigma/\sqrt{n}$. Here, $\sigma$ is the standard deviation of the sample and $n$ is the number of samples. The 95% confidence interval was calculated by median bootstrapping with 10,000 random samples from the distribution. Biological replicates are experiments conducted on different days with different larvae and reagents. We used the Kruskal-Wallis test (https://uk.mathworks.com/help/stats/kruskalwallis.html) to compare several samples, followed by a Dunn-Sidak post hoc test.

## Solution of the two-state master equation

We assumed that MTs can either grow or shrink, and each of these two states ($g$ and $s$ in short) has a probability distribution that depends on MT length $l$ and time $t$ ($p_g(l,t)$ and $p_s(l,t)$). MTs can furthermore stop growing and start shrinking with rate $f_g = 1/t_g$ ($t_g$ being the average MT growth time) and stop shrinking to start growing with rate $f_s = 1/t_s$ ($t_s$ being the average MT shrinkage time). Furthermore, MTs are assumed to grow with velocity $v_g$ and shrink with velocity $v_s$, while they are in the growing or shrinking state, respectively. Writing this as a master equation yields:

$$\frac{d}{dt} p_s(l,t) = f_g p_g(l,t) - f_s p_s(l,t)$$

$$\frac{d}{dt} p_g(l,t) = f_s p_s(l,t) - f_g p_g(l,t)$$

This equation is a two-state master equation that equates the rate change of a probability to be in one state (i.e., $dp_s/dt$) to the outflow ($-f_s p_s$, i.e., the likelihood of a shrinking MT to start growing) and the inflow ($f_g p_g$, i.e., the likelihood of a growing MT to start shrinking) into that state. With $dp/dt = \partial l/\partial t \partial p/\partial l = v \partial p/\partial l$, we can write:

$$\frac{\partial}{\partial t} p_s(l,t) = f_g p_g(l,t) - f_s p_s(l,t) + v_s \frac{\partial}{\partial l} p_s(l,t)$$

$$\frac{\partial}{\partial t} p_g(l,t) = f_s p_s(l,t) - f_g p_g(l,t) - v_g \frac{\partial}{\partial l} p_g(l,t)$$

To solve this set of partial differential equations, consider the following Fourier transformation of $p_s(l,t)$ and $p_g(l,t)$:

$$p_{s,g}(l,t) = \int dk \, d\omega \, e^{i\omega t - ikl} \widetilde{p}_{s,g}(\omega,k)$$

Substituting in the two-state master equation yields:

$$0 = \int dk \, d\omega \, e^{i\omega t - ikx} \left[ \left( i\omega + f_g - ikv_g \right) \widetilde{p}_g(\omega,k) - f_s \widetilde{p}_s(\omega,k) \right]$$

$$0 = \int dk \, d\omega \, e^{i\omega t - ikx} \left[ \left( i\omega + f_s + ikv_s \right) \widetilde{p}_s(\omega,k) - f_g \widetilde{p}_g(\omega,k) \right]$$

which can be written as a matrix equation:

$$0 = \begin{pmatrix} i\omega + f_g - ikv_g & -f_s \\ -f_g & i\omega + f_s + ikv_s \end{pmatrix} \begin{pmatrix} \widetilde{p}_g(\omega,k) \\ \widetilde{p}_s(\omega,k) \end{pmatrix}$$

This equation only has non-zero solutions for $\widetilde{p}_g$ and $\widetilde{p}_s$ if the matrix determinant is equal to zero:

$$0 = \det \begin{pmatrix} i\omega + f_g - ikv_g & -f_s \\ -f_g & i\omega + f_s + ikv_s \end{pmatrix} = \left( i\omega + f_g - ikv_g \right)\left( i\omega + f_s + ikv_s \right) + f_g f_s$$

This equation can be written as a dispersion relation:

$$\omega(k) = \overbrace{\left( \frac{f_s}{f_s + f_g} v_g - \frac{f_g}{f_s + f_g} v_s \right)}^{=\bar{v}} k + i \overbrace{\frac{f_s f_g (v_g + v_s)^2}{(f_g + f_s)^3}}^{=\overline{D}} k^2 + O(k^3) = \bar{v}k + i\, \overline{D}\, k^2 + O(k^3)$$

For large times $t$ both $\omega$ and $k$ are small so that we can drop terms of the order of $k^3$. The dispersion relation is then the same as for a diffusion advection process with drift velocity $v$ and diffusion coefficient $D$. For $v > 0$ the system will evolve like a diffusion advection process in which MTs would have no average length so that they will become as long as the system allows, that is, their growth is 'unbounded'.

For $v < 0$, MTs will exhibit an average length that depends on their dynamic parameters which can be calculated as follows: For large times, the overall probability to find an MT with length $l$ at time $t$ ($p(l,t) = p_g(l,t) + p_s(l,t)$) can be approximated by a modified diffusion-advection equation:

$$\frac{\partial}{\partial t}p\left(l,t\right) = D\frac{\partial^2}{\partial l^2}p\left(l,t\right) + v \vee \frac{\partial}{\partial l}p\left(l,t\right)$$

The stationary state $\frac{\partial}{\partial t}p\left(l,t\right) = 0$ is thus found by:

$$0 = \frac{\partial^2}{\partial l^2}p\left(l\right) + \frac{|v|}{D}\frac{\partial}{\partial l}p\left(l\right)$$

The general solution to this partial differential equation is:

$$p(l) = \frac{|\bar{v}|}{D}e^{-\frac{\bar{v}}{D}l}$$

For $p\left(l,t\right)$ to be normalizable: $C_2 = 0$ and $C_1 = \left(\frac{|v|}{D}\right)^2$. So that:

$$p\left(l\right) = \frac{|\bar{v}|}{D}e^{-\frac{\bar{v}}{D}l}$$

This equation is a two-state master equation that equates the rate change of a probability to be in one state (i.e., $dp_s/dt$) to the outflow ($-f_s p_s$, i.e., the likelihood of a shrinking MT to start growing) and the inflow ($f_g p_g$, i.e., the likelihood of a growing MT to start shrinking) into that state. With $dp/dt = \partial l/\partial t \partial p/\partial l = v\partial p/\partial l$, we can write:

$$\frac{\partial}{\partial t}p_s\left(l,t\right) = f_g p_g\left(l,t\right) - f_s p_s\left(l,t\right) + v_s\frac{\partial}{\partial l}p_s\left(l,t\right)$$

$$\frac{\partial}{\partial t}p_g\left(l,t\right) = f_s p_s\left(l,t\right) - f_g p_g\left(l,t\right) - v_g\frac{\partial}{\partial l}p_g\left(l,t\right)$$

Finally, one can calculate the average MT length $l_{MT}$ as the expectation value of the length:

$$l_{MT} \equiv \langle l \rangle = \int_0^\infty dl\, l\, p\left(l\right) = \frac{D}{|v|} = \frac{f_s f_g\left(v_g + v_s\right)^2}{\left(f_g + f_s\right)^2\left(v_s f_g - v_g f_s\right)}$$

For $v_s v_g \approx 1$, $f_s f_g \approx 1$, and $d = v/f$ the quadratic terms can be Taylor expanded to yield:

$$l_{MT} \approx \frac{v_g v_s}{\left(v_s f_g - v_g f_s\right)} = \frac{d_g d_s}{d_s - d_g}$$

## Analytical fit to estimate MT growth per cycle based on immunostainings

The p150 fluorescence profiles were calculated as described in the section on 'p150 antibody staining'. Subsequently, an exponential function, $p150\left(x\right) = b + e^{-s\left(x - x_0\right)}$, was fitted to the first 12 bins of the data (corresponds to up until 120 µm from the tip). Here, $b$, $s$, and $x_0$ are fitting parameters with units: [$b$]=AU, [$s$]=1/µm, [$x_0$]=µm. Next, we assumed that $d_g\left(x\right) = Ap150\left(x\right)^\alpha$, that is, MT growth per cycle is a simple power law in the p150 fluorescence intensity. $A$ and $\alpha$ are fitting parameters and their values are shown in *Figure 4—figure supplement 4* The expected growth length per cycle for an MT that starts growing at position $x$ towards (1) or away (–1) from the cell body can be approximated as:

$$d_g\left(x, sign\right) = 0.5\left(d_g\left(signd_g\left(x\right) + x\right) + d_g\left(x\right)\right)$$

Here, we assumed that the average length added to an MT during growth is the mean between the average expected growth length at start and end position of the MT plus-end. This function was subsequently fitted to the growth length data presented in *Figure 1F*. To do so, the experimental data was first binned in bins of size 10 µm (like the staining data). Then, we calculated the integral of $d_g\left(x, sign\right)$ over each bin for each direction of growth and minimized the squared difference to the experimental results by varying $A$ and $\alpha$. Note that we assumed that MTs that grow way from the tip have to be at least 4 µm away from it (average MT length; *Yu and Baas, 1994*) and that MTs that grow into the tip may penetrate it by 2 µm.

## MT sliding simulations

Details of the simulation can be found in *Jakobs et al., 2020*. We here present a brief description that focusses on the novel way in which new MTs are added during the simulation. MTs were arranged with their long axis along the x-axis of a Cartesian coordinate system and their centres on a hexagonal lattice in the y-z plane. For simplicity all MTs were assumed to have the same length, $l_{MT}$ = 4 µm. The inter-MT spacing in the y-z plane (~30 nm) was assumed to allow individual molecular motors (here, cytoplasmic dynein) to intervene between adjacent filaments and cross-link them with their respective 'cargo' or 'walking' domains. The simulation was initialized with 10 randomly oriented MTs that were randomly distributed on a hexagonal lattice of length 6 µm. New MTs were added to the system depending on the chosen nucleation model:

1. Sliding only: MTs were added at random locations with random orientation every 1100 s (~18 min). The time was optimized to yield axons of approximately the same length as cultured ones.
2. Sliding and templating: MTs were added at random locations every 1100 s. The likelihood of being plus-end-out was calculated by counting the number and orientation of MTs at the location (the centre of the MT) in which the MT is added. Then, the number of plus-end-out MTs was divided by the total number of MTs to calculate the probability of getting a plus-end-out MT. Finally, a random number is drawn between 0 and 1 to determine the orientation of the added MT.
3. Templating only: Same as above except that the directionality of the movement of molecular motors along the MTs was eliminated. When MT overlaps became occupied with motors, the motors' gliding direction towards the plus-end or the –end of the MTs was chosen at random. Note that this is a highly artificial setting that solely removes the sorting effect of MT sliding.
4. Sliding and unbounded growth: A random location along the axon was chosen and a random MT orientation (50/50 plus-end-out/minus-end-out) introduced every 435 s. As not every MT nucleated in this model, the rate of influx was selected to be higher to enable the same axon growth behaviour. Subsequently, we calculated the likelihood of exhibiting unbounded growth for an MT with the randomly selected orientation and location. To do so, we first calculated the average added length per growth cycle in 10 µm bins (distance from the axon tip and separately for plus-end-out and minus-end-out MTs) for each axon in the dataset presented in *Figure 1A–G*. For each bin we then queried whether growth was bounded (added length below 2.2 µm) or unbounded. The likelihood of unbounded growth was calculated for each bin by counting the number of axons that exhibited unbounded growth in the bin and dividing that number by all axons. Subsequently, two exponential functions were fitted to the plus-end-out and minus-end-out MT data respectively to determine a function that gives the likelihood of unbounded growth for plus-end-out and minus-end-out MTs as a function of distance from the axon tip. Finally, the random location and the predetermined orientation were used to look up the likelihood of unbounded growth and the MT was assumed to have nucleated successfully when a randomly drawn number [0,1] was smaller than that likelihood.
5. Unbounded growth only: Same as above but molecular motors were again assumed to not have a preferred direction as in 3.
6. Sliding, templating, and unbounded growth: A random location along the axon was chosen every 435 s and its orientation likelihood calculated as in 2. Subsequently, the unbounded growth likelihood was calculated as in 3. MTs only successfully entered the system if exhibiting unbounded growth.

MTs that were neighbours on the y-z plane and overlapping along the x-axis were cross-linked by cytoplasmic dynein. For simplicity and due to the tight packing of MTs in the bundle, only motion in parallel to the x-axis was considered. MT velocities were determined by solving a set of force balance equations that characterize dynein interaction with the MTs, as detailed in *Jakobs et al., 2020*. Furthermore, the left boundary was a leaky spring; MTs that moved into the left boundary were subject to a force of 50 pN/µm and were able to leave the axon with a fixed rate per MT (0.00024/s).

The rate was adjusted to lead to axons of 50 µm in length after approximately 24 hr simulation time. The right boundary was a constant force of 50 pN as described in *Jakobs et al., 2020*.

Axons were simulated for 50,001 iterations (~28 hr) and all results averaged over 50 separate simulations. Simulation parameters were as follows:

| Symbol | Description | Value | Reasoning |
|---|---|---|---|
| $\chi$ | Fraction of overlapping MTs that are cross linked | 1 | We previously explored how changing $\chi$ affects MT sliding (*Jakobs et al., 2020*; *Jakobs et al., 2015*). In this manuscript we simply wanted to explore the effect of different MT addition models on MT orientation in which we fixed the value at 1. |
| $\lambda$ | Number of motors bound in an overlapping region (#/µm) | 5 | We quantified (by eye) the number of MT cross-links in EM images of axons (*Hirokawa et al., 2010*) which was approximately 5 per 1000 nm. |
| $IMT$ | MT lengths (µm) | 4 µm | Average MT lengths in axons measured in *Yu and Baas, 1994*. |
| $\xi$ | Drag coefficient of the axoplasm | 1 pN s/µm$^2$ | Same coefficient used in *Oelz et al., 2018*. |
| $f_s$ | Dynein stall force | 1.4 pN | Same coefficient used in *Oelz et al., 2018*. |
| $v_0$ | Dynein free velocity | 0.86 µm/s | Same coefficient used in *Oelz et al., 2018*. |
| $dt$ | Simulation timestep per iteration | 2 s | As we showed previously (*Jakobs et al., 2020*; *Jakobs et al., 2015*), this value is a good choice to ensure smooth movements of MTs during the simulation. |

## Acknowledgements

We would like to thank Eva Pillai, Dennis Bray, Michael Takla, Kevin Chalut, and Melissa Rolls for inspiring discussions and proofreading, Andreas Prokop and Cristina Melero for teaching *Drosophila* dissection techniques, and Sarah Bray, Dmitry Nashchekin, Daniel St Johnston, and Nick Brown for providing *Drosophila* strains, laboratory space and a great atmosphere to work in. The authors acknowledge funding from the Wellcome Trust (PhD studentship 109145/Z/15/Z to MAHJ), the UK Biotechnology and Biological Sciences Research Council (Research Grant BB/N006402/1 to KF), the European Research Council (Consolidator Award 772426 to KF), and the Alexander von Humboldt Foundation (Alexander von Humboldt Professorship to KF).

## Additional information

### Competing interests

Maximilian AH Jakobs: MAHJ and KF are shareholders of deepMirror (https://deepmirror.ai), a company that, amongst other products, sells custom3 interfaces of the freeware KymoButler. Kristian Franze: KF is shareholders of deepMirror (https://deepmirror.ai), a company that, amongst other products, sells custom interfaces of the freeware KymoButler used in this study. The other author declares that no competing interests exist.

### Funding

| Funder | Grant reference number | Author |
|---|---|---|
| Wellcome Trust | PhD studentship 109145/Z/15/Z | Maximilian AH Jakobs |
| Biotechnology and Biological Sciences Research Council | Research Grant BB/N006402/1 | Kristian Franze |
| European Research Council | Consolidator Award 772426 | Kristian Franze |

| Funder | Grant reference number | Author |
| --- | --- | --- |
| Alexander von Humboldt-Stiftung | Alexander von Humboldt Professorship | Kristian Franze |

The funders had no role in study design, data collection and interpretation, or the decision to submit the work for publication. For the purpose of Open Access, the authors have applied a CC BY public copyright license to any Author Accepted Manuscript version arising from this submission.

## Author contributions

Maximilian AH Jakobs, Conceptualization, Data curation, Software, Formal analysis, Funding acquisition, Investigation, Visualization, Methodology, Writing – original draft, Writing – review and editing; Assaf Zemel, Resources, Software, Supervision, Methodology, Writing – review and editing; Kristian Franze, Conceptualization, Supervision, Funding acquisition, Visualization, Writing – original draft, Writing – review and editing

## Author ORCIDs

Maximilian AH Jakobs http://orcid.org/0000-0002-0879-7937
Assaf Zemel http://orcid.org/0000-0002-6816-3303
Kristian Franze http://orcid.org/0000-0002-8425-7297

## Decision letter and Author response

Decision letter https://doi.org/10.7554/eLife.77608.sa1
Author response https://doi.org/10.7554/eLife.77608.sa2

# Additional files

## Supplementary files

• Transparent reporting form

## Data availability

The software used in this study is freely available, a Gitlab link is provided in the manuscript. Data files can be found on biostudies and accessed via: https://www.ebi.ac.uk/biostudies/studies/S-BIAD547.

The following dataset was generated:

| Author(s) | Year | Dataset title | Dataset URL | Database and Identifier |
| --- | --- | --- | --- | --- |
| Franze K | 2022 | Drosophila primary neuron microtubule imaging data | https://www.ebi.ac.uk/biostudies/bioimages/studies/S-BIAD547 | EMBL-EBI, S-BIAD547 |

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
