## [Editor Report]

How axons form and maintain uniformly plus-end-out microtubules is an essential question in neuronal cell biology. Franze and colleagues used solid imaging and modeling approaches to provide important insights into the mechanisms controlling microtubule polarity in cultured *Drosophila* axons. They conclude that reduced catastrophe of the plus-end-out microtubules in the axon tip is critical for preferential plus-end-out microtubule growth and establishing the uniform microtubule polarity.

---

## [Decision Letter]

**Decision letter after peer review:**

[Editors’ note: the authors submitted for reconsideration following the decision after peer review. What follows is the decision letter after the first round of review.]

Thank you for submitting the paper "Unrestrained growth of correctly oriented microtubules instructs axonal microtubule orientation" for consideration by *eLife*. Your article has been reviewed by 3 peer reviewers, and the evaluation has been overseen by a Reviewing Editor and a Senior Editor. The following individual involved in review of your submission has agreed to reveal their identity: Andreas Prokop (Reviewer #1).

Comments to the Authors:

We are sorry to say that, after consultation with the reviewers, we have decided that this work will not be considered further for publication by *eLife*.

While all the reviewers found the proposed concepts and the model interesting and potentially a good addition to the field, two out of three reviewers thought that the experimental part of the manuscript is too preliminary and the model requires more experiments to be properly tested. Such additional experiments would go beyond the scope of what we normally ask for in an *eLife* revision, and therefore, we return the paper to you. However, if you would able to thoroughly address all the reviewer comments, in particular by providing additional experimental data to strengthen your conclusions, we will be prepared to consider a new submission of this manuscript which we will try to send to the same reviewers.

*Reviewer #1 (Recommendations for the authors):*

In this manuscript the authors address potential mechanisms through which microtubules in developing axons gradually achieve uniform plus end-out orientation. Combining cell biological experiments and computational modelling they propose a combinatorial model in which (1) anterograde transport generates a gradient of polymerisation-promoting factors at the axon tip, thus biassing +end elongation near growth cones, (2) dynein-mediated retrograde transport removes 'wrongly' oriented microtubules, and (3) further bias is generated by directional nucleation of new microtubules in the axon shaft.

The ideas proposed in this manuscript are attractive and some of the experiments, such as the double-labelling of Eb1::GFP with Jupiter::mCherry, are exciting. The dynein-related functions provide new explanations for reported phenotypes that loss of Dhc causes polarity effects in axons. However, the conclusions are not sufficiently supported by data. More experiments should have been provided to support claims, especially when considering that the fly neurons used are highly amenable to efficient experimentation. Key experiments would have been to use loss of dynein heavy chain to confirm the p150 link to retrograde transport in this context, to use loss of Grip/augmin proteins (http://www.ncbi.nlm.nih.gov/pubmed/23132930) to test the importance of directional nucleation in the axon shaft, and direct tests of the intensities of polymerisation-relevant proteins (e.g. Eb1 or Msps; https://doi.org/10.1371/journal.pgen.1009647) which would be expected to be weaker in proximal than distal axons. The use of p150 is not suitable to this end because, as the authors state, it is "mainly known for its role in the dynactin complex" (l.208); the claim "we here found that an enrichment of microtubule growth-promoting proteins at the advancing axon tip leads to a transition of microtubule growth from a bounded to an unbounded state" (l.193) is therefore not supported by the data. Also, in their model dynein is mainly suggested to be involved in retrograde transport of minus end-out microtubules not in polymerisation. As a further possibility to support the key claim, velocity of Eb1 comets could have been provided as another measure of polymerisation efficiency that relates to amounts of Eb1 at plus ends (https://doi.org/10.7554/*eLife*.51992). Finally, the manuscript has not made clear to me why retrograde polymerisation events in certain axon segments are shorter than anterograde events based on the model provided; the authors should have given some potential explanations. Can they exclude that these are minus end polymerisation events which get increasingly toned down through minus end stabilisation? These points would need to be discussed.

l.37. 'RNAs, proteins and organelles'.

l.42-45: slightly confusing; rearrange and assign references clearly.

l.49: MT symmetry is not an ideal term and misleading; 'mixed orientation' might be clearer?

l.57ff. dynein has been shown to affect axonal polarity (https://doi.org/10.1038/ncb1777), and the pioneer work by Peter Baas and colleagues to explain such a phenomenon should be mentioned here.

l.57ff. A further polarity factor is Shot: http://doi.org/10.1242/dev.00319 further discussed here: http://doi.org/10.1016/j.semcdb.2017.05.019

End of Intro: It would be great to have a concluding sentence that leads over to the results.

l.68: Stepanova used EB3; it might be better to cite (one of) the first *Drosophila* papers using Eb1 in culture and also providing info on directionality: https://doi.org/10.1002/dneu.20762

l.72: since the assumption is that Eb1 comets mark only +ends, it would be better to rename '-end out' into '+end in' which would be easier to grasp.

l.71ff. the idea of minus end polymerisation cannot be excluded and needs to be incorporated into the thinking. What if minus ends become more stable over time? Furthermore, polymerisation velocity is a function of Eb1 amounts (https://doi.org/10.7554/*eLife*.51992, https://doi.org/10.1371/journal.pgen.1009647) and has not been considered.

l.79: you mean density not lifetime here?

l.81: replace 'growth' by 'polymerisation' and reserve 'growth' for axon growth?

l.83ff. This statement requires data that plot lifetime as a function of directionality and age, which is not provided; Figure 1H needs to be cited but, only provides positional correlation.

l.106: 'catastrophe or pause events'.

l.106ff/Figure 2: beautiful experiment. It would be helpful to see polymerisation as well as shrinkage plotted for + end-out and -in MTs, rather than one integrated plot in E.

l.133: also have a look at https://doi.org/10.1371/journal.pgen.1009647 for relevant factors.

l.138ff. The localisation of p150 is not what I would expect of dynein in axons; since p150 is a CAP-Gly protein that can track + tips (https://doi.org/10.1038/nrm2369), I wonder whether the dots along the axon are at polymerising plus ends? Where is that prominent accumulation of p150 localised in GCs relative to MTs? It reminds of similar accumulations seen for CLIP170/190 (https://doi.org/10.1091/mbc.E14-06-108). Higher resolution images would be very important here.

l.141: Based on what property of p150 do you build the model and make the claim that it should influence MT orientation? I cannot follow your reasoning here. Is this a dynein-related function? I assume so when considering the model proposed below. If so, independent experiments with loss of dynein heavy chain would be very important.

l.174: Here it would be easy to validate your computational findings with real experimental data using Grip/augmin genes for which adequate tools are available (http://www.ncbi.nlm.nih.gov/pubmed/23132930).

l.189: see this review of mechanisms regulating MT polymerisation: http://doi.org/10.1016/j.brainresbull.2016.08.006

l.208ff.: the link to dynein should have been explored experimentally.

l.240: here the work by Peter Baas should be cited who was (one of) the first to propose the idea of selective minus end-out MT transport as a mechanism to maintain polarity.

l.488: remove Viki Allan as last author.

*Reviewer #2 (Recommendations for the authors):*

This manuscript explores how microtubule orientation in axons transitions from slightly biased towards plus-end out to almost completely plus-end out. By measuring microtubule growth episodes in axons of cultured fly neurons, the authors demonstrate that growth events are longer near the axon tip. The authors then hypothesized that these longer events might tip the balance from bounded growth to unbounded growth, which would explain why axonal microtubules become increasingly oriented plus-end out as the axon develops further and grows longer. To test this, the authors measured shrinkage lengths and demonstrate that growth indeed only exceeds shrinkage in the case of plus-end out growing microtubules near the axon tip. Consistently, after changing the balance between growth and shrinkage using nocodazole axons no longer became exclusively plus-end out.

The authors then searched for factors that could promote microtubule growth near axon tips. They focused on p150, a microtubule binder that is enriched at the axon tip, and found that experimental manipulations of p150 resulted in altered microtubule orientations. Finally, the authors developed a numerical simulations that examine how various microtubule orienting and sorting approaches (i.e. templated nucleation, motor-based sliding and tip-promoted growth) can result in uniform microtubule orientations along the axonal length.

The key innovation in this manuscript is the careful measurement of both microtubule growth and shrinkages event in cultured fly axons. Through this, the authors could conclude that plus-end out oriented microtubules close to the axon tip are in the unbounded growth regime, whereas other microtubules are not. This selective advantage will ensure an increasingly uniform orientation as the axon growth longer. This is an interesting and important insight into the establishment of oriented microtubule arrays. Although the exact mechanism of this local growth promotion awaits further exploration, the author's proposal that tip-enriched microtubule regulators could establish this provides interesting leads for further research.

Comments:

– In Supplementary Figure 4, the authors fit their growth data to a model that included the experimentally observed distribution of p150. It should be emphasized that they added two free fitting parameters to achieve this (A and \α), which, by their definition as amplitude and power, should be able to stretch any exponential decay function into another. It would be helpful if the authors could also report the values of A and \α that were found in the fitting procedure. This could give more insights into the estimated spatial range of p150 activity in comparison to its distribution. Furthermore, it would be nice if the authors could explain how they think that p150 mediates growth enhancement while being tip localized. Do they think p150 is anchored or soluble? If anchored, how can it promote growth?

– The choice to focus on p150 is somewhat enigmatic and could be better introduced. The Mouhamiam paper cited focused on a role from p150 in transport initiation. The fact that p150 disruption or kinesin-1 knockdown result in altered microtubule organization does not provide direct support for their model, given the direct roles of dynein and kinesin-1 in microtubule organization through sliding that have been proposed. The authors discuss these points in the discussion, but it would be nice if they could elaborate a bit on how they think p150 promotes microtubule growth at the axon tip.

– It is unclear how the authors model the interaction between microtubules and the plasma membrane. Often microtubules that grow against a barrier undergo catastrophes and that would counteract the promoted growth of plus-end out oriented microtubules. The methods section has a statement about forces at the ends of the axon, but this point was not entirely clear and should be discussed.

– The discussion on TRIM46 appears inaccurate in several places. For example, as far as I know there is no evidence TRIM46 is locally synthesized at axon tips, as suggested in line 136. Furthermore, TRIM46 likely plays a role in a fourth mechanism contributing to uniform polarity, namely the selective stabilization of properly oriented microtubules through selective parallel bundling. This would need to be discussed.

– In principle, the finite pool of tubulin dimers could also help in diluting out the minus-end out oriented microtubules if the growth of plus-end out microtubules is specifically promoted. Modeling this would be a new effort, but the authors could at least discuss this.*Reviewer #3 (Recommendations for the authors):*

Summary

Jacobs et al. explore the question of how axons develop a nearly uniform array of microtubules (MTs) that point with their plus ends towards the growth cone during development. Using experimental analysis of *Drosophila* CNS larval neurons grown in vitro, they document that, like N2A and *Xenopus* neurons, there is a higher density of Eb1 approaching the growth cone. Additionally, they report that along axons, MTs that point with their plus ends toward the growth cone have a longer growth length than MTs points to the cell body. Using Jupiter-mcherry to track MT growth and shrinkage, they find the average length of shrinkage is 2.2 μm and use this as input, along with the growth lengths, in a mathematical model that estimates average MT length based on these parameters. To test the model, neurons are treated with nocodazole and a high concentration of NaCl to perturb microtubule assembly and cellular osmolarity. Both treatments lead to a significant decrease in the growth length of MTs and a decrease in the percentage of MTs pointing towards the growth cone. Building on the prior observation that p150glued promotes MT assembly in neurons, they confirm a gradient in p150 localization along axons and that p150-RNAi decreases the concentration of p150 in growth cones. Analyzing the effects on MT assembly, they report that reducing p150 levels leads to decreases in microtubule growth length and organization. Using the experimental data, they create what they call an unbound growth model. The core idea is that MTs that grow longer distances than they shrink will have lengths that approach infinity over time, i.e., growth is unbounded. In contrast, MTs that shrink more than the growth distance (but never wholly depolarize) will have a finite length (i.e., bounded growth). Bringing these experimental observations together with prior models describing how uniform arrays of MTs are generated in axons, they construct a computational model. The model suggests that combining the effects of dynein mediated MT sliding, unbounded growth, and augmin based MT templating leads to highly organized arrays of MT where MTs along the length of the axon essentially all point towards the growth cone.

Major strengths and weaknesses:

The major strength of the manuscript is consideration of how differences in MT assembly as function of position along the axon and MT orientation might contribute to the generation of a uniform array of MTs in the axon.

There are three main weaknesses. The first is whether ‘unbounded’ MT growth occurs in axons needs to be clarified. The second is whether the difference in MT assembly as a function of orientation is ‘real’ or an artifact. And finally, the modelling seems to set up MT sliding as a strawman to make a case for the importance of differential MT assembly. Nonetheless, I think these points can be addressed by reanalyzing data, extending the modelling analysis, and a more careful discussion of the results.

Appraisal of work

A significant point of the manuscript is that MTs oriented with their plus ends towards the growth cone in the last ten microns of the axon undergo ‘unbounded’ growth because the distance they polymerize is greater than the distance they depolymerize. In contrast, MTs in the axon and those pointing to the cell body have bounded growth because the reverse occurs. Nonetheless, the data indicates the length of MT shrinkage is greater than MT growth in all cases. If MTs only undergo bounded growth, please rewrite sections of the manuscript that suggest otherwise. Alternatively, explain how unbound growth might be occurring in the context of the model.

There is a concern that the difference in MT growth length as a function of MT orientation may be an artifact of how growth length was measured. To start, it is assumed that MT growth increases toward the growth cone. Thus, one would expect some difference, be it large or small, in MT growth across a ten-micron section (i.e., the distance assessed in the experiments). In general, I would assume that MTs growing towards the growth cone will tend to have their plus ends, on average, in the front half of the ten-micron section, and MTs growing towards the cell body will have the opposite distribution. This raises the concern that the growth rate of MTs going opposite directions may be a function where the MTs plus ends are located, on average, in the ten-micron section; instead of their orientation. This concern could be addressed by systematically reducing the bin size to determine if the difference in growth length disappears or converges to some finite non-zero value as the bin size becomes small. If there are other ways to address this, please do so.

Some may view the modelling of the effects on MT sliding as presenting a strawman argument to claim that MT sliding is insufficient to explain the clearance of MTs pointing +end towards the cell body. To clarify, the computational model is founded on a simulation where MT sliding contributes to the attainment of a uniform MT distribution along the axon. With the model as presented, the parameters for MTs directed towards the cell body at the axon/cell body boundary have been set such that these MTs accumulate in the axon instead of moving into the cell body. Based on a prior manuscript, one would expect that a minor adjustment to one parameter describing the behavior at the boundary will lead to a situation where MT sliding generates axonal arrays of MT that point almost uniformly towards the growth cone. Thus, it seems artificial to suggest the sliding model fails, and ‘unbounded growth’ is needed to generate a uniform array of MTs. On this point, the modelling section would be improved if the effects of each aspect (i.e., ‘unbounded growth,’ MT templating, and MT sliding) were first addressed separately and then combined into a unified model.

[Editors’ note: further revisions were suggested prior to acceptance, as described below.]

Thank you for resubmitting your work entitled “Unrestrained growth of correctly oriented microtubules instructs axonal microtubule orientation” for further consideration by *eLife*. Your revised article has been evaluated by Anna Akhmanova (Senior Editor) and a Reviewing Editor.

The manuscript has been improved but there are some remaining issues that need to be addressed, as outlined below. We encourage you to consider these points when you submit a revised manuscript. The editors will make decisions about your revised manuscript without sending it back to the reviewers.

*Reviewer #1 (Recommendations for the authors):*

The paper has now become much clearer, and the change in terminology has had a huge impact on getting the point across. I strongly recommend its publication.

I read the manuscript very carefully and have a number of comments to strengthen the final publication and avoid unnecessary criticism. All aspects concern changes to the text, and only my comment on l.160 (Noc, osmo+) might trigger the addition of some data which might be readily available anyway.

Detailed comments:

Abstract: in my view, the abstract could be simplified along the lines of “we found that anterogradely polymerising MTs prevail through a mechanism that protects their +ends from catastrophes whereas retrograde MTs are unprotected and show limited growth.” Then the p150-dependent mechanisms could be explained. Also, it would be good to name the three mechanisms that constitute the final model.

l. 45: Can we say that the minus end is stabilised? If anchored to γ-tub this might be true, but minus ends of most axonal MTs regulated by CAMSAP-katanin seem to be in a dynamic steady-state (http://doi.org/10.1016/j.devcel.2014.01.001 and http://doi.org/10.1016/j.str.2017.12.017)

l.71/73: format error of references.

l.93: is growth necessarily ended through catastrophe? I envisage pauses that turn into plus end stabilisation through, for example, CLASP (http://doi.org/10.1101/gad.17015911) – see further comments below.

l.101-103.: Just a thought: throughout the manuscript, it might be better to spell out plus end and minus end? the symbols are easily missed, and the minus looks sometimes like a hyphen. In general the +end out/-end out nomenclature is confusing: in the example in lines 101/2 you leave open as to whether you speak about plus or minus end polymerisation in the case of minus end out MTs. In line 103 it is not clear whether you speak of retrograde plus tips within the 10micron distal stretch. -- It might help to come up with a clearer nomenclature. Suggestion: "plus ends of anterograde MTs added significantly more …. than plus ends of retrograde MTs." The term antero/retrograde orientation of MTs is currently not used, but could be introduced here to make description of this phenomenon easier in the future. If not happy with this solution speak at least of plus end-in/out rather than minus end-out and plus end-out. In Figure 1, this could also be clarified by drawing blue/red arrows into the axon. Note that Figure 1L should be changed to 1K.

l.108. capitalise *Drosophila*.

l.112: add our recent paper showing the Eb1 amount/velocity correlation also in axons (https://doi.org/10.1371/journal.pgen.1009647)

l.113. In figure 1 you cannot yet speak about catastrophe rates, since you do not know whether MTs pause or depolymerise. You can only speak about the length of polymerisation bouts. This is different from Figure 2 onwards. If in experiments underpinning Figure 2 you never see pauses, this would be an argument to 'assume' that catastrophe is causing termination of polymerisation.

p.7/8: for the non-mathematical trained reader it might be helpful to define the un-/bounded terms a bit better. I understand the principal idea but wonder why you set the boundary between un-/bounded at 2 micron which looks a bit arbitrary/artificial unless there is a mathematical rationale. Is it necessary to introduce the un-/bounded terms? Can one not simply speak of smaller/larger average distance of polymerisation?

l.137: … in axons of larval primary neurons?

l.148: Does "any MT further away from the tip" refer to both antero- and retrograde MTs? This would mean that the statements "a higher chance of survival for +end out MTs" in line 150 would have to add "within 10 micron distance from the tip"

l.160. In my naive non-mathematician view, the interpretation of the experiment with Noc and osmo+ only makes sense if the effect is stronger on anterograde than on retrograde. Are there no/negligible effects on retrograde MTs? No data are shown for retrograde MTs.

l.173: please, differentiate the molecular function through which stabilisation is achieved: p150 is an anti-collapse factor, CRMP a promoter of polymerisation and TRIM46 a MT cross-linker. Why tubulin heterodimers are mentioned here is not clear to me, and TRIM46 is enriched at the AIS rather than axon tip, but it has been shown to promote polarity (see also my comments on line 234).

l.182: fly neurons in culture rarely display dendrites, and larval cultures usually fail to differentiate into synaptic maturity (see section 4.1 in http://dx.doi.org/10.1007/978-1-61779-830-6_10), which would have to be checked with synaptic markers in your culture system. From the simplicity of the projection, I doubt that the side branch shown is a dendrite, which would also be expected to arise from the cell body (https://doi.org/10.1016/j.ydbio.2005.09.026). Furthermore, it is well-established that MTs in invertebrate dendrites are almost completely plus end-in (Melissa Rolls' reviews). Overall, I do not think that the dendrite aspect is important for your manuscript and should better be left out?

l.187. You would have to specify that your prediction concerns vertebrate dendrites, whereas *Drosophila* dendrites have polar MTs in their majority/all pointing towards the cell body (Melissa Rolls' reviews)

l.200. Did you do fly or larval cultures? heterozygous mutant *Drosophila* larvae?

l.232: please, briefly provide the key parameters on which the 2020 model is built: was it bases on MT sliding? Please, also refer to the model by Rao et al. (https://doi.org/10.1016/j.celrep.2017.05.064) highlighting potential deviations (might be an issue also for the discussion)

l.234ff. The van Beuningen reference is inadequate here since their model is more based on AIS-dependent functions of TRIM46 through unknown mechanisms potentially involving selective transport. The better reference might be Rao et al. (https://doi.org/10.1016/j.celrep.2017.05.064) who propose directional stabilisation of polar MTs through TRIM46 versus unhindered retrograde transport.

To my knowledge the closest fly homologue to TRIM46 is TRIM9 (although not convincingly close) which stabilises polymerisation in an orientation-dependent fashion in fly dendrites (https://doi.org/10.1242/jcs.258437) – also present in sensory axons (https://doi.org/10.1016/j.jcg.2010.12.004)

l.243/245: correct to 'catastrophe-inhibiting protein'

l.291: as pointed out earlier, TRIM46 is enriched proximally, and there is, to my knowledge, no obvious bias for distal Eb1 enrichment apart from the fact that there might be more polymerisation events in growth cones. Better examples would be Tau which has long been known to enrich distally.

l.293. As argued earlier, the dendrite data are very shaky and should better be taken out.

Discussion: a summary image illustrating the various mechanism and how they contribute would be very helpful to reach a wider audience.

*Reviewer #2 (Recommendations for the authors):*

The revised manuscript by Jacobs et al., entitled "Unrestrained growth of correctly oriented microtubules instructs axonal microtubule orientation" is improved, yet important concerns remain.

The main point of the manuscript is to understand how axons acquire a uniform MT orientation. From the title and abstract, the main conclusion would appear to be that unrestrained MT growth plays a vital role in this process. Nonetheless, the modeling in the current version of the manuscript suggests that unbounded growth makes a minor contribution to the establishment of axonal MT polarity (e.g., Sup Figure 8C). This disconnect between what is claimed (e.g., "we confirmed that the enhanced growth of +end out microtubules is critical for achieving uniform microtubule orientation.") and what is demonstrated by the experimental data and modeling is problematic. Additionally, the manipulations used to perturb MT dynamics (i.e., p150, kinesin, and nocodazole) are also known to alter dynein activity. Since it is well accepted that dynein plays a role in establishing MT orientation, much of the data can be viewed as confirmation that dynein is essential for establishing axonal microtubule orientation.

The following paragraphs summarize the main findings in the manuscript, point out issues with the current version, and suggest approaches the authors could consider in terms of aligning the data and modeling to improve the manuscript.

The key experimental findings and issues are:

1. +end out microtubules are less likely to undergo catastrophe near the advancing axon tip, leading to unbounded MT growth.

The critical issue here is that the modeling, as it stands, suggests unbounded growth is not very important for establishing a uniform microtubule array. Thus, while this is an interesting result, its importance for establishing MT orientation is unclear.

2. Decreasing MT growth with NOC or osm+ disrupts MT orientation

While this could be interpreted to mean that modulating MT dynamics alters MT orientation, because nocodazole is known to disrupt dynein, the effect of nocodazole on MT orientation could be occurring through a change in dynein activity rather than a change in MT assembly.

3. Disrupting p150 function reduces MT growth (dg um/cycle) in the last 10 μm of the axon and reduces MT orientation.

As noted in the previous review, p150 influences dynein activity. Thus, the effects on MT orientation could be occurring through a change in dynein activity rather than MT dynamics.

4. Disruption of dynein disrupts MT orientation, decreases MT growth (i.e., dg) in the last ten microns of the axon, but increases dg along the axon.

The observation that disruption of dynein alters MT dynamics is interesting. Still, the observation that dg decreases near the growth cone and rises along the axon makes it challenging to interpret the results.

5. Disruption of kinesin (khc-RNAi) reduces p150 levels in the growth cone, disrupts MT orientation, decreases MT growth (i.e., dg) in the last ten microns of the axon, but increases MT growth along the axon.

Disruption of khc is known to disrupt dynein function in *Drosophila* neurons (Pilling et al., 2006, paper out of Saxton's lab). While the effects may be occurring by reducing p150 levels in the growth cone, a more straightforward explanation is that the disruption in MT orientation is occurring because disruption of kinesin disrupts dynein. Additionally, the decrease in dg near the growth cone and increase in dg along the axon is hard to interpret (Sup Figure 7F).

6. Based on these observations and previous studies, the manuscript proposes a model for how axons achieve a uniform MT orientation. The modeling suggests that unbounded MT assembly works with MT sliding and templating to establish a uniform MT array.

There are multiple issues with the modeling. The first is that the extent of rapid Stop and Go MT sliding in the experimental data set is not analyzed. From studies in *Xenopus* neurons from Popov's group (Ma et al., 2004 Cur Bio) and fly neurons (Gelfand's group) rapid sliding of MTs is exceedingly rare after neurite initiation. Based on this data, it seems likely that if the velocity distribution of comet motion were examined, essentially none of the comets would be moving at the rate of dynein/kinesin motors (i.e., ~ 1 um/sec). In turn, it follows a careful analysis of the Jupiter mcherry data might reveal that only tiny fraction, if any, of the MTs move via rapid MT sliding. I would suggest carefully looking at the experimental data and updating the model so it accurately reflects the observed extent of MT sliding.

Sup Figure 8C suggests that unbounded MT growth alone does not contribute to microtubule orientation. Comparison of 8A, 8C, and 8E suggests unbounded growth modestly increases MT orientation in the proximal axon when sliding is included in the model. If one takes the model's output at face value, the verbal arguments in the manuscript claiming unbounded growth is essential for establishing MT orientation are undermined. I would suggest either accepting that the model indicates unbounded growth is relatively unimportant for establishing MT orientation or rethinking the model.

It's unclear why augmin is given such a prominent role in the model and the discussion. I found this distracting.

In conclusion, there are two significant issues. The first is that manipulations designed to alter MT dynamics, either directly or by reducing the localization of p150 to the growth cone (i.e., p150, noc, kinesin), also disrupt dynein, which is well established in being essential for establishing MT orientation. As a result, the experimental data do not appear to show a definitive role for MT dynamics separate from dynein activity. The second is that the modeling suggests MT dynamics make a relatively minor contribution towards establishing MT orientation. In light of this, approaches that could be used to strengthen the manuscript would be to conduct experiments that more directly demonstrate the importance of MT dynamics in this process and to revise the model as outlined above.

*Reviewer #3 (Recommendations for the authors):*

The authors have made additional efforts to address the comments raised by the reviewers, which has led to various improvements of their manuscript.

In Supplemental Figure 5, the authors now report the fitted values and found a value for \α of 3.62. Given that d_g(x)=A*p150(x)^\α, this indicates that the gradient of microtubule behavior has a 4x steeper spatial decay than the gradient of p150. This could be the case, but it is not apparent from the stainings and profile plots of p150 they show in Figure 4C and S7. Based on Supplementary Figure 2, it seems that the p150 gradient is actually steeper. Furthermore, I had hoped that the authors would have added some reflection on how a gradient of p150 can result in a gradient of MT dynamics with a 4-fold different length scale. To me this is not entirely evident, but it could be due to a non-linear response of MT dynamics to p150 concentration.

Unfortunately, the authors didn't try to address the dynamics of p150 to assess to which extent was diffuse versus anchored (e.g. using FRAP), and therefore the exact interplay between p150 and MTs at the axon tip remains unclear. Nonetheless, the manuscript does provide a more coarser insight into how uniform MT arrays could be established by local modulation of MT dynamics.

---

## [Author Response]

[Editors’ note: the authors resubmitted a revised version of the paper for consideration. What follows is the authors’ response to the first round of review.]

Reviewer #1 (Recommendations for the authors):In this manuscript the authors address potential mechanisms through which microtubules in developing axons gradually achieve uniform plus end-out orientation. Combining cell biological experiments and computational modelling they propose a combinatorial model in which (1) anterograde transport generates a gradient of polymerisation-promoting factors at the axon tip, thus biassing +end elongation near growth cones, (2) dynein-mediated retrograde transport removes 'wrongly' oriented microtubules, and (3) further bias is generated by directional nucleation of new microtubules in the axon shaft.The ideas proposed in this manuscript are attractive and some of the experiments, such as the double-labelling of Eb1::GFP with Jupiter::mCherry, are exciting. The dynein-related functions provide new explanations for reported phenotypes that loss of Dhc causes polarity effects in axons.

We would like to thank the reviewer for this positive assessment of our study.

However, the conclusions are not sufficiently supported by data. More experiments should have been provided to support claims, especially when considering that the fly neurons used are highly amenable to efficient experimentation. Key experiments would have been to use loss of dynein heavy chain to confirm the p150 link to retrograde transport in this context.

We added new experimental results on dynein downregulation in Figure S6 as suggested by the reviewer, highlighting similarities between p150 and dynein downregulation.

To use loss of Grip/augmin proteins (http://www.ncbi.nlm.nih.gov/pubmed/23132930) to test the importance of directional nucleation in the axon shaft.

While we agree with the reviewer that this is an interesting experiment, we would like to refer to recent studies investigating the role of augmin in regulating microtubule orientation (Nguyen et al., 2014; Sánchez-Huertas et al., 2016). We have added more detailed information about these experiments to the revised version of our manuscript (l.57ff).

And direct tests of the intensities of polymerisation-relevant proteins (e.g. Eb1 or Msps; https://doi.org/10.1371/journal.pgen.1009647) which would be expected to be weaker in proximal than distal axons.

In the revised manuscript, we directly measured the intensity of EB1-GFP fluorescence as a function of position within the axon. In contrast to the reviewer’s expectation, EB1-GFP fluorescence is slightly lower at the axon tip (i.e., distally) compared to the rest of the axon (data shown in the new Figure S1E). Additionally, we would like to highlight that microtubule growth speeds were uniform along the axon, while catastrophe frequencies decreased towards the axon tip (new Figure S1A-C), indicating that catastrophe but not polymerisation is affected by protein gradients at the axon tip.

The use of p150 is not suitable to this end because, as the authors state, it is "mainly known for its role in the dynactin complex" (l.208); the claim "we here found that an enrichment of microtubule growth-promoting proteins at the advancing axon tip leads to a transition of microtubule growth from a bounded to an unbounded state" (l.193) is therefore not supported by the data.

In the revised manuscript, we directly measured the intensity of EB1-GFP fluorescence as a function of position within the axon. In contrast to the reviewer’s expectation, EB1-GFP fluorescence is slightly lower at the axon tip (i.e., distally) compared to the rest of the axon (data shown in the new Figure S1E). Additionally, we would like to highlight that microtubule growth speeds were uniform along the axon, while catastrophe frequencies decreased towards the axon tip (new Figure S1A-C), indicating that catastrophe but not polymerisation is affected by protein gradients at the axon tip.

Also, in their model dynein is mainly suggested to be involved in retrograde transport of minus end-out microtubules not in polymerisation.

We agree that dynein is mostly involved in microtubule sliding and not in MT polymerisation. However, we here focused on dynactin rather on dynein, which contains p150. As stated above, p150 is a microtubule anti-catastrophe factor independently of the dynactin complex. Of course, dynactin may also bind to dynein. As shown in the new figure S6, dynein downregulation affects MT catastrophe rates (and microtubule orientation) as well, likely at least partly due to its interactions with dynactin.

As a further possibility to support the key claim, velocity of Eb1 comets could have been provided as another measure of polymerisation efficiency that relates to amounts of Eb1 at plus ends (https://doi.org/10.7554/eLife.51992).

We added Eb1 velocities to Figures S1, S3, S4, S6, and S7 as suggested by the reviewer. As shown in these figures, MT polymerisation velocities are constant along the axon. We would like to emphasize that we did not mean to state that MT polymerisation changes along the axon. Instead, catastrophe rates are affected, which then leads to increases in the average microtubule growth lengths per growth cycle.

Finally, the manuscript has not made clear to me why retrograde polymerisation events in certain axon segments are shorter than anterograde events based on the model provided; the authors should have given some potential explanations. Can they exclude that these are minus end polymerisation events which get increasingly toned down through minus end stabilisation? These points would need to be discussed.

A microtubule that grows towards a gradient of microtubule growth promoting protein / anti catastrophe factor will grow longer than a microtubule that grows away from that gradient. We demonstrated this by calculating the differences in microtubule growth lengths per cycle for +end out and -end out microtubules in a growth promoting protein gradient as shown in Figure S5 of the original manuscript. The calculation showed that the observed gradient is steep enough to induce different growth behaviours for +end out and -end out microtubules.

While we cannot fully exclude -end polymerisation events, we would like to point out that microtubule -end polymerisation is several orders of magnitude slower than +end polymerisation (Strothman et al., 2019). Our data shows that both cell body-directed and axon tip-directed polymerisation events exhibit similar speeds along the axon, strongly suggesting that all polymerisation events observed are +end polymerisation events (new Figure S1c). To address this point, we now explain our simulation in more detail, and we discuss end polymerisation (*l.434ff*)

l.37. 'RNAs, proteins and organelles'.

Changed as suggested.

l.42-45: slightly confusing; rearrange and assign references clearly.

Rearranged and added references to first sentence of paragraph.

l.49: MT symmetry is not an ideal term and misleading; 'mixed orientation' might be clearer?

Changed as suggested.

l.57ff. dynein has been shown to affect axonal polarity (https://doi.org/10.1038/ncb1777), and the pioneer work by Peter Baas and colleagues to explain such a phenomenon should be mentioned here.

We had already mentioned the effect of dynein on axonal polarity at the end of the paragraph in the original manuscript and cited Peter Baas. We now cite the reference earlier for clarity (*l.65*).

l.57ff. A further polarity factor is Shot: http://doi.org/10.1242/dev.00319 further discussed here: http://doi.org/10.1016/j.semcdb.2017.05.019

Shot is a microtubule/actin cytoskeletal linker protein and a neuronal polarity factor, as mentioned by the reviewer. However, while neuronal polarity and microtubule orientation are certainly related, Shot has, to the best of our knowledge, not been shown to regulate microtubule orientation. Hence, we would prefer not to mention it in l.54ff of the revised manuscript, were we discuss microtubule orientation.

End of Intro: It would be great to have a concluding sentence that leads over to the results.

We changed the concluding sentence as suggested.

l.68: Stepanova used EB3; it might be better to cite (one of) the first *Drosophila* papers using Eb1 in culture and also providing info on directionality: https://doi.org/10.1002/dneu.20762

We thank the reviewer for the citation, which we added to the manuscript (*l.89*).

l.72: since the assumption is that Eb1 comets mark only +ends, it would be better to rename '-end out' into '+end in' which would be easier to grasp.

As most previous publications appear to use the expressions +end out and -end out (e.g., (Stone et al., 2008; Wang et al., 2019)), we would prefer to keep this notation.

l.71ff. the idea of minus end polymerisation cannot be excluded and needs to be incorporated into the thinking. What if minus ends become more stable over time?

-end microtubule polymerisation is much slower than +end polymerisation (Strothman et al., 2019). Thus, it is very unlikely that retrograde polymerisation events observed in our study correspond to -end polymerisation. Furthermore, MT -ends in *Drosophila* have been shown to be capped by Patronin (del Castillo et al., 2015), a *Drosophila* homologue of CAMSAP3, which has been shown to prevent polymerisation (Hendershott and Vale, 2014). It is thus unlikely that -ends polymerise much. Exploring -end stability during development is certainly an interesting avenue for further research, but it is beyond the scope of the current study.

Furthermore, polymerisation velocity is a function of Eb1 amounts (https://doi.org/10.7554/eLife.51992, https://doi.org/10.1371/journal.pgen.1009647) and has not been considered.

As we now show, polymerisation velocities are uniform along the axon (Figure S1C), and EB1 comet lengths are unchanged as well (Figure S1F), suggesting that polymerisation velocities of MTs do not contribute to the different growth behaviours of +end out and -end out microtubules observed in our study.

l.79: you mean density not lifetime here?

We are not sure which sentence the reviewer refers to. We did not use the word ‘lifetime’ anywhere in the manuscript.

l.81: replace 'growth' by 'polymerisation' and reserve 'growth' for axon growth?

We were careful to not use ‘polymerisation’ as it is often used to describe growth velocity. As this manuscript mostly uses the added length per growth cycle as a parameter, “growth” actually describes better what we want to say. We use polymerisation only when talking about polymerisation velocities. To address this concern, we rephrased some statements to be clear which parameters we refer to.

l.83ff. This statement requires data that plot lifetime as a function of directionality and age, which is not provided; Figure 1H needs to be cited but, only provides positional correlation.

We thank the reviewer for the comment and reworded the statement to indicate a positional correlation *(l.104ff)*.

l.106: 'catastrophe or pause events'.

Done.

l.106ff/Figure 2: beautiful experiment. It would be helpful to see polymerisation as well as shrinkage plotted for + end-out and -in MTs, rather than one integrated plot in E.

We thank the reviewer for his positive assessment of this experiment, which unfortunately was a very low throughput experiment. Of all the 47 axons in which we could identify shrinkage events, only two exhibited events for -end out microtubules. In total, there were 6 events of which 5 came from a single axon. Hence, our data were statistically not robust enough to separate +end out and -end out shrinkage. However, separating the existing data on +end out and -end out growth yielded very similar d_s_(+end out) = 2.03 [1.80, 2.26] µm/cycle and d_s_(-end out) = 2.05 [0.61, 3.50] µm/cycle (medians and 95% confidence intervals), suggesting that pooling the data is reasonable.

l.133: also have a look at https://doi.org/10.1371/journal.pgen.1009647 for relevant factors.

This paper shows that XMAP215/Msps promotes microtubule polymerisation. However, to the best of our knowledge, there is no data showing an enrichment of this polymerase at axon tips. Since the paragraph the reviewer is referring to focuses on the enrichment of proteins at axon tips, we would prefer not to mention it here.

l.138ff. The localisation of p150 is not what I would expect of dynein in axons; since p150 is a CAP-Gly protein that can track + tips (https://doi.org/10.1038/nrm2369), I wonder whether the dots along the axon are at polymerising plus ends? Where is that prominent accumulation of p150 localised in GCs relative to MTs? It reminds of similar accumulations seen for CLIP170/190 (https://doi.org/10.1091/mbc.E14-06-108). Higher resolution images would be very important here.

The fixation protocol that was required to stain p150 did not conserve microtubule tips. Therefore, we were unable to perform colocalization studies. However, as the reviewer pointed out, there is substantial evidence in the literature that p150 does track +tips, and overexpression of fluorescently tagged p150 leads to similar “comet” like structures as EB1GFP overexpression.

l.141: Based on what property of p150 do you build the model and make the claim that it should influence MT orientation? I cannot follow your reasoning here. Is this a dynein-related function? I assume so when considering the model proposed below. If so, independent experiments with loss of dynein heavy chain would be very important.

This seems to be a misunderstanding. p150 is a known anti-catastrophe factor, which is why we explored it here in more detail. This function is not dynein-related (Lazarus et al., 2013). To address this important point and avoid confusion, we now explain our motivation to look at p150 in more detail in the Results part (*l.167ff*), we added plots of catastrophe rates to all experiments, and we extended the description of our model (*l.325ff*), in which gradients of microtubule anti-catastrophe factors accumulate at axon tips, thus promoting growth of +end out microtubules growing into the gradient but not of -end out microtubules growing out of the gradients.

l.174: Here it would be easy to validate your computational findings with real experimental data using Grip/augmin genes for which adequate tools are available (http://www.ncbi.nlm.nih.gov/pubmed/23132930).

The importance of augmin for uniform plus end-out orientation of MTs has already been shown in the literature (Nguyen et al., 2014; Sánchez-Huertas et al., 2016). However, how augmin contributes to MT sorting is still unclear. Our simulations suggest that, while augmin is required to establish MT polarity, neither augmin-based nucleation alone nor in conjunction with dynein-based MT sliding lead to the experimentally observed uniform MT orientation along axons. We feel that perturbing augmin expression as in the publications mentioned above would, while being easy to do, not directly test the model and thus not lead to further insights.

l.189: see this review of mechanisms regulating MT polymerisation: http://doi.org/10.1016/j.brainresbull.2016.08.006

We thank the reviewer for this review article, which we now cite in *l.171*.

l.208ff.: the link to dynein should have been explored experimentally.

We have added experimental perturbations of dynein as suggested. Our new data show how dynein downregulation leads to similar effects on microtubule growth as p150 downregulation, in line with our model (Figure S6).

l.240: here the work by Peter Baas should be cited who was (one of) the first to propose the idea of selective minus end-out MT transport as a mechanism to maintain polarity.

We thank the reviewer for spotting this and added the relevant reference in *l.333*.

l.488: remove Viki Allan as last author.

Done.

Reviewer #2 (Recommendations for the authors):This manuscript explores how microtubule orientation in axons transitions from slightly biased towards plus-end out to almost completely plus-end out. By measuring microtubule growth episodes in axons of cultured fly neurons, the authors demonstrate that growth events are longer near the axon tip. The authors then hypothesized that these longer events might tip the balance from bounded growth to unbounded growth, which would explain why axonal microtubules become increasingly oriented plus-end out as the axon develops further and grows longer. To test this, the authors measured shrinkage lengths and demonstrate that growth indeed only exceeds shrinkage in the case of plus-end out growing microtubules near the axon tip. Consistently, after changing the balance between growth and shrinkage using nocodazole axons no longer became exclusively plus-end out.The authors then searched for factors that could promote microtubule growth near axon tips. They focused on p150, a microtubule binder that is enriched at the axon tip, and found that experimental manipulations of p150 resulted in altered microtubule orientations. Finally, the authors developed a numerical simulations that examine how various microtubule orienting and sorting approaches (i.e. templated nucleation, motor-based sliding and tip-promoted growth) can result in uniform microtubule orientations along the axonal length.The key innovation in this manuscript is the careful measurement of both microtubule growth and shrinkages event in cultured fly axons. Through this, the authors could conclude that plus-end out oriented microtubules close to the axon tip are in the unbounded growth regime, whereas other microtubules are not. This selective advantage will ensure an increasingly uniform orientation as the axon growth longer. This is an interesting and important insight into the establishment of oriented microtubule arrays. Although the exact mechanism of this local growth promotion awaits further exploration, the author's proposal that tip-enriched microtubule regulators could establish this provides interesting leads for further research.

We would like to thank the reviewer for this positive assessment of our study.

Comments:– In Supplementary Figure 4, the authors fit their growth data to a model that included the experimentally observed distribution of p150. It should be emphasized that they added two free fitting parameters to achieve this (A and \α), which, by their definition as amplitude and power, should be able to stretch any exponential decay function into another. It would be helpful if the authors could also report the values of A and \α that were found in the fitting procedure. This could give more insights into the estimated spatial range of p150 activity in comparison to its distribution.

We thank the reviewer for raising this excellent point. As the reviewer pointed out, we used an exponential model which assumed that the average growth length during a given growth cycle is proportional to A*p150^α. It is important to highlight that we simultaneously fitted this function to both the microtubules growing away from the cell body and those growing towards it. Hence, we found a single pair of A = 0.9 and α = 3.6 which led to similar differences in MT growth as observed experimentally. This was only possible because the experimentally observed p150 gradient was strong enough to lead to differences in the average growth length of a microtubule. We added this discussion and the values to the legend of Figure S5 of the revised manuscript.

Furthermore, it would be nice if the authors could explain how they think that p150 mediates growth enhancement while being tip localized. Do they think p150 is anchored or soluble? If anchored, how can it promote growth?

As explained above and in much more detail in the revised manuscript, p150 mediates MT growth enhancement through decreasing catastrophe rates. The mechanism by which p150 promotes growth is subject of ongoing research, and at this point we can only speculate whether p150 is soluble or anchored. Previous work assumed p150 to be soluble and suggested that it uses its CAP-Gly domains to hold tubulin subunits in close vicinity of each other, thus making the growing microtubule tip more stable and preventing catastrophes (Lazarus et al., 2013).

– The choice to focus on p150 is somewhat enigmatic and could be better introduced. The Mouhamiam paper cited focused on a role from p150 in transport initiation. The fact that p150 disruption or kinesin-1 knockdown result in altered microtubule organization does not provide direct support for their model, given the direct roles of dynein and kinesin-1 in microtubule organization through sliding that have been proposed. The authors discuss these points in the discussion, but it would be nice if they could elaborate a bit on how they think p150 promotes microtubule growth at the axon tip.

As mentioned in our response to reviewer 1, p150 is a known microtubule anti-catastrophe factor, which is why we explored it here in more detail. This function is not dynein-related (Lazarus et al., 2013). To address this important point, we now explain our motivation to look at p150 in more detail in the Results part (*l.167ff*), we show plots of catastrophe rates for all experiments, and we extended the description of our model, in which gradients of microtubule anti-catastrophe factors accumulate at axon tips, thus promoting growth of +end out microtubules growing into the gradient but not of -end out microtubules growing out of the gradients.

– It is unclear how the authors model the interaction between microtubules and the plasma membrane. Often microtubules that grow against a barrier undergo catastrophes and that would counteract the promoted growth of plus-end out oriented microtubules. The methods section has a statement about forces at the ends of the axon, but this point was not entirely clear and should be discussed.

Elegant work by Marileen Dogterom and others has shown that, in vitro, catastrophe frequencies are increased when MTs polymerize into an obstacle. We see a similar effect in *Drosophila* neurons. As shown in Author response image 1, MTs do exhibit decreased growth lengths and increased catastrophe frequencies in the very distal region of the axon when growing into the tip (which is counter-balanced by higher concentrations of anti-catastrophe factors such as p150). We did not explicitly model the interaction between MTs and the plasma membrane, which results in higher catastrophe rates for MTs at the tip. However, since our simulation directly utilises the experimentally measured growth lengths, we indirectly account for decreased growth into the tip.

**Author response image 1. sa2fig1:** Growth length per cycle and catastrophe frequency vs distance from axon tip. Here we replotted the data shown in SFigure 1A-B with a bin size of 2µm instead of 10µm. Differences in d_g_ and f_g_ exists also at smaller bin sizes.

– The discussion on TRIM46 appears inaccurate in several places. For example, as far as I know there is no evidence TRIM46 is locally synthesized at axon tips, as suggested in line 136. Furthermore, TRIM46 likely plays a role in a fourth mechanism contributing to uniform polarity, namely the selective stabilization of properly oriented microtubules through selective parallel bundling. This would need to be discussed.

We apologise for the confusion cause by our wording. We phrased the discussion about TRIM46 more carefully now, clarified its known functions in *l.59ff* of the revised manuscript, and highlighted the simulation results that pertain to that function (*l.334ff*).

– In principle, the finite pool of tubulin dimers could also help in diluting out the minus-end out oriented microtubules if the growth of plus-end out microtubules is specifically promoted. Modeling this would be a new effort, but the authors could at least discuss this.

We thank the reviewer for this intriguing thought, which we added it to the discussion (l.329ff).

Reviewer #3 ( (Recommendations for the authors)):[...]There are three main weaknesses. The first is whether ‘unbounded’ MT growth occurs in axons needs to be clarified. The second is whether the difference in MT assembly as a function of orientation is ‘real’ or an artifact. And finally, the modelling seems to set up MT sliding as a strawman to make a case for the importance of differential MT assembly. Nonetheless, I think these points can be addressed by reanalyzing data, extending the modelling analysis, and a more careful discussion of the results.

We thank the reviewer for the critical reading of our manuscript and would like to refer to our detailed responses to these concerns below.

Appraisal of workA significant point of the manuscript is that MTs oriented with their plus ends towards the growth cone in the last ten microns of the axon undergo ‘unbounded’ growth because the distance they polymerize is greater than the distance they depolymerize. In contrast, MTs in the axon and those pointing to the cell body have bounded growth because the reverse occurs. Nonetheless, the data indicates the length of MT shrinkage is greater than MT growth in all cases. If MTs only undergo bounded growth, please rewrite sections of the manuscript that suggest otherwise. Alternatively, explain how unbound growth might be occurring in the context of the model.

In the old Figure 2F the reviewer is referring to, we showed median ± lower and upper quantiles for +end out and -end out MTs. Only a fraction of +end out MTs near the growth cone entered the regime of unbounded growth. We would like to point out that, even if the median growth length is shorter than the median shrinkage length (averaged over many axons), many microtubules will still exhibit unbounded growth even though the average microtubule does not. Those are the ones that survive in the long term and give rise to the uniform MT orientation in the axon.

However, we understand how the plot we showed could be misleading. In order to avoid confusion, we now use bootstrapped median values and their 95% confidence intervals as the reviewer suggested further below and replotted our data accordingly. Our new Figure 2 is more intuitive and demonstrates more clearly that (only) +end out MT growth lies above the threshold for unbounded growth.

There is a concern that the difference in MT growth length as a function of MT orientation may be an artifact of how growth length was measured. To start, it is assumed that MT growth increases toward the growth cone. Thus, one would expect some difference, be it large or small, in MT growth across a ten-micron section (i.e., the distance assessed in the experiments). In general, I would assume that MTs growing towards the growth cone will tend to have their plus ends, on average, in the front half of the ten-micron section, and MTs growing towards the cell body will have the opposite distribution. This raises the concern that the growth rate of MTs going opposite directions may be a function where the MTs plus ends are located, on average, in the ten-micron section; instead of their orientation. This concern could be addressed by systematically reducing the bin size to determine if the difference in growth length disappears or converges to some finite non-zero value as the bin size becomes small. If there are other ways to address this, please do so.

We thank the reviewer for this insightful comment. We now added a supplementary figure that highlights the distribution of growth displacements at bin size 10 µm (Figure S1). Also, please find a plot with bin size 2 µm in Author response image 1 (bootstrapped medians +/- 95% confidence intervals). As seen in that plot, even for small bin sizes of 2 µm, oppositely oriented MTs still exhibit growth differences within 4 µm from the axon tip, making it unlikely that the observed differences in d_g_ and f_g_ arose solely from the location of microtubules within a bin.

Some may view the modelling of the effects on MT sliding as presenting a strawman argument to claim that MT sliding is insufficient to explain the clearance of MTs pointing +end towards the cell body. To clarify, the computational model is founded on a simulation where MT sliding contributes to the attainment of a uniform MT distribution along the axon. With the model as presented, the parameters for MTs directed towards the cell body at the axon/cell body boundary have been set such that these MTs accumulate in the axon instead of moving into the cell body. Based on a prior manuscript, one would expect that a minor adjustment to one parameter describing the behavior at the boundary will lead to a situation where MT sliding generates axonal arrays of MT that point almost uniformly towards the growth cone. Thus, it seems artificial to suggest the sliding model fails, and ‘unbounded growth’ is needed to generate a uniform array of MTs.

We thank the reviewer for giving us the opportunity to address this seeming contradiction. We now clarified where the deficiency of the “sliding only” mechanism lies and how the biased growth mechanism contributes to the establishment of uniform MT orientation along the axon. First, we added a new panel to Figure 5 (5D) showing experimental plots of the MT orientation profile along the axon as a function of developmental time. In agreement with previous literature, young axons exhibited a graded MT orientation profile, with an increasing number of -end out microtubules towards the soma. This orientation profile became flatter for axons that grew longer, and microtubules were oriented uniformly along the entire length of the axon within 24 hours.

In our simulations which included dynein-mediated sliding of MTs only, we also obtained a graded MT orientation profile with dominating -end out orientation in the proximal axon. The failure of this model was that the scaled MT orientation profile (x-axis scaled by the axon length) attained a steady–state shape in which a fixed fraction of the axon remained with mixed MT orientation. This means that the size of the proximal region with mixed MT orientation continued to increase in proportion to the length of the growing axon. The reason for this observation was that, as the axon became longer, more numerous –end-out MTs accumulated at the axon entry from across the axon. In our previous paper we have shown that one can reduce the extent of this region by applying a stronger load on the bundle or by reducing the motor connectivity to the MTs, however, we could not eliminate the mixed region entirely while preserving the ability of the axons to grow. Thus, the sliding-only mechanism failed to explain the long-term polarization of the axon.

While working on our previous manuscript, we also tried to increase the “permeability” of the boundary. However, when more MTs left the axon and disappeared, MT bundles did not grow any longer and collapsed, suggesting that some boundary is required to build up stable MT bundles. And even if there was no boundary between axon and soma, MTs would not just disappear in the soma but rather continue growing and sliding. In fact, MT bundles in *Drosophila* neurons tend to continue towards the other side of the cell body, which in itself presents an impenetrable boundary (Author response image 2). Either way, MTs encounter a boundary, and the location of this boundary will only determine at which side of the soma the region with mixed MT orientations starts. This region would always span a significant fraction of the proximal axon, particularly for longer axons, if sliding was the only mechanism employed to bias MT orientations.

**Author response image 2. sa2fig2:** Expansion microscopy image of *D. melanogaster* neuron. We stained α-tubulin (yellow) and expanded the sample ~5x. Individual microtubule bundles enter the cell body from the axon and continue to the other side, suggesting that -end out microtubules leaving the axon do not disappear but rather move until hitting a boundary at the opposite side of the soma.

The two other mechanisms incorporated in our current simulation, namely, the localized bias of MT nucleation (templating), and our newly found mechanism of biased +end MT growth at the axon tip, both significantly improved the long-term behaviour of the MT orientation profile. However, when only the templating mechanism was added, there was the “risk” that axons with an inverted (-end out) MT orientation would form if incidentally the initial fraction of –end out MTs in the axon was higher. When simulating axons with templating only (SI Figure 8B), or with sliding+templating only (SI Figure 8D), we thus occasionally obtained bundles with inverted polarity. Biased growth of +end out MTs at the tip seems crucial to prevent this situation. Finally, when combining the three mechanisms, we obtained MT orientation profiles that closely resembled our experimental data (see new Figure 5). We have added this information to our discussion of the simulation results in the main text and the supplementary information.

On this point, the modelling section would be improved if the effects of each aspect (i.e., ‘unbounded growth,’ MT templating, and MT sliding) were first addressed separately and then combined into a unified model.

The new Figure S8 now contains simulations of templating and unbounded growth alone.

[Editors’ note: what follows is the authors’ response to the second round of review.]

Reviewer #1 (Recommendations for the authors):The paper has now become much clearer, and the change in terminology has had a huge impact on getting the point across. I strongly recommend its publication.

We thank the reviewer for this very positive evaluation of our revised manuscript.

I read the manuscript very carefully and have a number of comments to strengthen the final publication and avoid unnecessary criticism. All aspects concern changes to the text, and only my comment on l.160 (Noc, osmo+) might trigger the addition of some data which might be readily available anyway.Detailed comments:Abstract: in my view, the abstract could be simplified along the lines of “we found that anterogradely polymerising MTs prevail through a mechanism that protects their +ends from catastrophes whereas retrograde MTs are unprotected and show limited growth.” Then the p150-dependent mechanisms could be explained. Also, it would be good to name the three mechanisms that constitute the final model.

We changed the abstract to reflect the reviewer’s suggestions.

l. 45: Can we say that the minus end is stabilised? If anchored to γ-tub this might be true, but minus ends of most axonal MTs regulated by CAMSAP-katanin seem to be in a dynamic steady-state (http://doi.org/10.1016/j.devcel.2014.01.001 and http://doi.org/10.1016/j.str.2017.12.017)

This is a good point. Changed to: “more stable”.

l.71/73: format error of references.

We apologise for this oversight. Reference format corrected.

l.93: is growth necessarily ended through catastrophe? I envisage pauses that turn into plus end stabilisation through, for example, CLASP (http://doi.org/10.1101/gad.17015911) – see further comments below.

We added a reference to Figure 2 to this section, in which we show that most MTs indeed started shrinking after EB1-GFP disappearance, indicating that our approximation is reasonable.

l.101-103.: Just a thought: throughout the manuscript, it might be better to spell out plus end and minus end? the symbols are easily missed, and the minus looks sometimes like a hyphen. In general the +end out/-end out nomenclature is confusing: in the example in lines 101/2 you leave open as to whether you speak about plus or minus end polymerisation in the case of minus end out MTs. In line 103 it is not clear whether you speak of retrograde plus tips within the 10micron distal stretch. -- It might help to come up with a clearer nomenclature. Suggestion: "plus ends of anterograde MTs added significantly more …. than plus ends of retrograde MTs." The term antero/retrograde orientation of MTs is currently not used, but could be introduced here to make description of this phenomenon easier in the future. If not happy with this solution speak at least of plus end-in/out rather than minus end-out and plus end-out. In Figure 1, this could also be clarified by drawing blue/red arrows into the axon. Note that Figure 1L should be changed to 1K.

We thank the reviewer for this helpful suggestion. In line with other published work (Akhmanova and Steinmetz, 2019; del Castillo et al., 2015), we changed +end out to plus-end-out throughout the manuscript. Additionally, we clarified the nomenclature in lines 76ff of the revised manuscript and added arrows to Figure 1.

l.112: add our recent paper showing the Eb1 amount/velocity correlation also in axons (https://doi.org/10.1371/journal.pgen.1009647)

Done (l.114).

l.113. In figure 1 you cannot yet speak about catastrophe rates, since you do not know whether MTs pause or depolymerise. You can only speak about the length of polymerisation bouts. This is different from Figure 2 onwards. If in experiments underpinning Figure 2 you never see pauses, this would be an argument to 'assume' that catastrophe is causing termination of polymerisation.

This is a good point. We added a sentence referencing Figure 2B-D in lines l.93ff of the revised manuscript, which states that most MTs in our experiments underwent catastrophes after EB1-GFP labelled polymerisation bouts.

p.7/8: for the non-mathematical trained reader it might be helpful to define the un-/bounded terms a bit better. I understand the principal idea but wonder why you set the boundary between un-/bounded at 2 micron which looks a bit arbitrary/artificial unless there is a mathematical rationale. Is it necessary to introduce the un-/bounded terms? Can one not simply speak of smaller/larger average distance of polymerisation?

Bounded and unbounded are standard terms used to describe functions: bounded means that the endpoints of a function are finite numbers, while functions are unbounded if the endpoint goes to infinity. To make this important point clear, we added a definition to the text (lines 133ff.).

The threshold is not arbitrary/artificial but based on our experimental data. As discussed in lines 146ff of the revised manuscript, MT growth is bounded when dg<ds due to equation (1). As shown in Figure 2E, ds=2.03 µm, so that the boundary between unbounded and bounded growth lies at approximately 2 µm

l.137: … in axons of larval primary neurons?

Yes. Changed as suggested.

l.148: Does "any MT further away from the tip" refer to both antero- and retrograde MTs? This would mean that the statements "a higher chance of survival for +end out MTs" in line 150 would have to add "within 10 micron distance from the tip"

The reviewer correctly pointed out that any MT further away from the tip than 10um exhibits similar growth to minus-end-out microtubules close to the axon tip, while only plus-end-out microtubules within 10 microns from the tip have a higher chance of unbounded growth. However, this still means that, on average, +end out MTs have a higher chance of survival. Thus, we would rather keep the statement as it is.

l.160. In my naive non-mathematician view, the interpretation of the experiment with Noc and osmo+ only makes sense if the effect is stonger on anterograde than on retrograde. Are there no/negligible effects on retrograde MTs? No data are shown for retrograde MTs.

The effect of both osmo+ and Noc on +end out MTs >10um from the tip was insignificant (Figure 3D). Similarly, the effect on -end out MTs was also insignificant (see Author response image 3). Thus, it appears as if both treatments have a baseline-dependent effect on MT growth, i.e., a stronger effect on MTs that exhibit higher growth and lower catastrophe frequencies.

**Author response image 3. sa2fig3:** Added MT lengths per growth cycle dg of minus-end-out MTs at the distalmost 10µm from the axon tip and further away for control (N = 107 axons from 5 biological replicates), nocodazole-treated axons (N = 116 axons from 3 biological replicates), and axons cultured in osmo+ medium (N = 30, 2 biological replicates). No significant differences were observed (p < 0.47 Kruskal Wallis test).

l.173: please, differentiate the molecular function through which stabilisation is achieved: p150 is an anti-collapse factor, CRMP a promoter of polymerisation and TRIM46 a MT cross-linker. Why tubulin heterodimers are mentioned here is not clear to me, and TRIM46 is enriched at the AIS rather than axon tip, but it has been shown to promote polarity (see also my comments on line 234).

We added further clarification to lines 179ff to highlight the functions through which stabilisation is achieved. We previously added tubulin heterodimers to our manuscript following a suggestion by reviewer #3. We liked this suggestion and would prefer to leave the reference in the manuscript, as a surplus of free tubulin may facilitate MT growth. As noted in lines 299ff of the revised manuscript, TRIM46 has been shown to accumulate at axon tips as well (Rao et al., 2017).

l.182: fly neurons in culture rarely display dendrites, and larval cultures usually fail to differentiate into synaptic maturity (see section 4.1 in http://dx.doi.org/10.1007/978-1-61779-830-6_10), which would have to be checked with synaptic markers in your culture system. From the simplicity of the projection, I doubt that the side branch shown is a dendrite, which would also be expected to arise from the cell body (https://doi.org/10.1016/j.ydbio.2005.09.026). Furthermore, it is well-established that MTs in invertebrate dendrites are almost completely plus end-in (Melissa Rolls' reviews). Overall, I do not think that the dendrite aspect is important for your manuscript and should better be left out?

We agree with the reviewer that the minor processes exhibited by fly neurons in our cultures are likely not yet fully functional dendrites as they do not exhibit the characteristic minus-end-out MT orientation observed in mature invertebrate dendrites in vivo. This, however, is the reason why we called them dendritic processes (rather than dendrites) throughout the paper, as they resembled dendrites and exhibited non-axonal MT orientation, i.e., mixed MT orientation. We now explicitly introduce them as such in l.189ff of the revised manuscript: ‘tips of immature ‘dendritic’ processes, whose MT orientation is mixed and thus resembles that of vertebrate and immature *Drosophila* dendrites (Hill et al. 2012)’. We feel that the data presented for these dendritic processes support our main finding – that selective stabilisation of +end out MTs near the axon tip is crucial for the development of a uniform +end out orientation of MTs in axons – and would thus prefer to keep the data.

l.187. You would have to specify that your prediction concerns vertebrate dendrites, whereas *Drosophila* dendrites have polar MTs in their majority/all pointing towards the cell body (Melissa Rolls' reviews)

MTs in the dendritic processes investigated in the present study are similar to those in vertebrate dendrites (as now clarified, l.189ff). They exhibit lower growth in the periphery and mixed overall orientation, thus corroborating an important connection between MT growth and orientation, which is what we intended to show in lines 189ff.

l.200. Did you do fly or larval cultures? heterozygous mutant *Drosophila* larvae?

Yes, we used heterozygous mutant larval neurons. We added missing information accordingly.

l.232: please, briefly provide the key parameters on which the 2020 model is built: was it bases on MT sliding? Please, also refer to the model by Rao et al. (https://doi.org/10.1016/j.celrep.2017.05.064) highlighting potential deviations (might be an issue also for the discussion)

We added further information on our previous model in lines 247ff. To keep our manuscript comprehensible, however, we would rather not want to discuss the model in the context of other models, which we already did extensively in our 2020 paper anyway. Additionally, the paper by Rao et al. (Rao et al., 2017) does not simulate MT-MT sliding but rather MT sliding along cortically attached dynein, and it also only calculates sliding velocities but not microtubule orientation. Hence, it would be a rather inadequate comparison in this part of the manuscript and the discussion.

l.234ff. The van Beuningen reference is inadequate here since their model is more based on AIS-dependent functions of TRIM46 through unknown mechanisms potentially involving selective transport. The better reference might be Rao et al. (https://doi.org/10.1016/j.celrep.2017.05.064) who propose directional stabilisation of polar MTs through TRIM46 versus unhindered retrograde transport.To my knowledge the closest fly homologue to TRIM46 is TRIM9 (although not convincingly close) which stabilises polymerisation in an orientation-dependent fashion in fly dendrites (https://doi.org/10.1242/jcs.258437) – also present in sensory axons (https://doi.org/10.1016/j.jcg.2010.12.004)

We thank the reviewer for this insightful comment and changed the references accordingly line 246.

l.243/245: correct to 'catastrophe-inhibiting protein'.

Done.

l.291: as pointed out earlier, TRIM46 is enriched proximally, and there is, to my knowledge, no obvious bias for distal Eb1 enrichment apart from the fact that there might be more polymerisation events in growth cones. Better examples would be Tau which has long been known to enrich distally.

As mentioned above and in the manuscript, (Rao et al., 2017) demonstrated that TRIM46 accumulates at axon tips (Figure 5A-B in the paper). Additionally, we cited two references which demonstrated that EB1-GFP is enriched at axonal tips (Ma et al., 2004; Morrison et al., 2002).

l.293. As argued earlier, the dendrite data are very shaky and should better be taken out.

We would like to reiterate that we do not think that these processes are real dendrites; we also don’t think that our data are shaky. In contrast, MTs in the dendritic processes exhibited reduced MT growth and mixed MT orientation, thus supporting our argument that MT growth is important for MT orientation.

Discussion: a summary image illustrating the various mechanism and how they contribute would be very helpful to reach a wider audience.

A summary of all discussed mechanisms is provided in Figure 5G.

Reviewer #2 (Recommendations for the authors):The revised manuscript by Jacobs et al., entitled "Unrestrained growth of correctly oriented microtubules instructs axonal microtubule orientation" is improved, yet important concerns remain.The main point of the manuscript is to understand how axons acquire a uniform MT orientation. From the title and abstract, the main conclusion would appear to be that unrestrained MT growth plays a vital role in this process. Nonetheless, the modeling in the current version of the manuscript suggests that unbounded growth makes a minor contribution to the establishment of axonal MT polarity (e.g., Sup Figure 8C). This disconnect between what is claimed (e.g., "we confirmed that the enhanced growth of +end out microtubules is critical for achieving uniform microtubule orientation.") and what is demonstrated by the experimental data and modeling is problematic.

Indeed, the results shown in Suppl. Figure 8C suggest that biased growth alone has a small effect on the orientation of MTs along the axon. Nevertheless, panels 8E and 8F show that, when this mechanism works in concert with dynein-mediated MT sliding and MT templating, its effect becomes significant enough to explain the gradual enrichment of the proximal axon with plus-end-out MTs, and the consequent development of a uniform MT orientation across the axon’s entire length.

This behaviour at the proximal end could not be explained by MT sliding alone (Suppl. Figure 8A) because the rate at which minus-end-out MTs accumulated in the proximal axon exceeded the rate at which these MTs got absorbed into the cell body. Both MT templating and biased growth of plus-end-out MTs at the axon tip are processes that dilute minus-end-out MTs in the axonal shaft, and thereby allow the MT array to eventually be cleared out from those minus-end-MTs by dynein-mediated sliding.

The reason for the small effect seen in Suppl. Figure 8C is that the MT growth-promoting factor is concentrated in a small region close to the axon tip, and its range of action thus diminishes in comparison to the axon length when the axon extends. That is, as the axon extends, a growing portion of its length continues to accumulate ill-oriented MTs. Sliding is therefore essential for clearing out the generated minus-end-out MTs, and this occurs much more efficiently in the presence of a bias at the growing tip that continually maintains the fraction of minus-end-out MTs small during growth.

Sliding alone, however, turns out to be insufficient for clearing the axonal shaft from accumulating ill-oriented MTs (only a graded polarity profile is formed as seen in Suppl. Figure 8A). The reason is that the longer the axon extends, the larger is the number of minus-end-out MTs generated across its length per unit time. This enlargement in the number of ill-oriented MTs is prevented by the biased MT polymerization at the axon tip. Consequently, the synergistic action of both mechanisms (or of the three mechanisms) significantly improves the resulting polarity of the axonal MT array.

Finally, we would like to note that our current simulations likely underestimate the effect of a growth promoting factor at the tip. The reason is that in our current simulations the actual polymerization dynamics of the MTs were not accounted for explicitly. Had these dynamics been taken into account, minus-end-out MTs would likely be shorter than plus-end-out ones and would thus be expelled from the axon much more quickly than they currently do. However, since appropriate modelling of the length distribution of MTs in the context of a growing axon requires many mechanisms to be accounted for (including MT polymerization / depolymerization dynamics, MT nucleation, MT severing, and the regulation of these processes by accessory proteins), we leave this challenge for a separate systematic investigation.

Additionally, the manipulations used to perturb MT dynamics (i.e., p150, kinesin, and nocodazole) are also known to alter dynein activity. Since it is well accepted that dynein plays a role in establishing MT orientation, much of the data can be viewed as confirmation that dynein is essential for establishing axonal microtubule orientation.

We respectfully disagree with the reviewer. According to literature (del Castillo et al., 2015; Rao et al., 2017), dynein contributes to microtubule orientation mostly through sliding. To the best of our knowledge, it remains unclear whether p150 is required for MT sliding, as mentioned in our discussion (see for example (Waterman-Storer et al., 1997) and (Tan et al., 2018)). Furthermore, we are not aware of any direct effect of Nocodazole on dynein-based MT sliding. In the literature, Nocodazole appears to disrupt dynein function mainly through dynein mislocalisation because of Nocodazole-mediated MT depolymerisation (Gerlitz et al., 2013). In our experiments, however, we used low Nocodazole doses and microtubules still polymerised (see Figure 3), so that it is unlikely that most of the dynein present becomes mislocalised, thus suggesting little effects on dynein function. Finally, while there might be some crosstalk between different mechanisms contributing to establishing uniform MT orientation in axons, our physical models (Figures 2 and S5) and simulations (Figures 5 and S8) clearly demonstrate a critical role of unbounded MT growth in orienting MTs – independent of the effect of dynein.

In the revised version of our manuscript, we extended the discussion of potential additional effects of our perturbations on dynein function and explain why it is a key contributor to the overall uniform axonal MT orientation (l.312ff). We furthermore extended our discussion explaining why dynein-based MT sliding is not sufficient to generate axons with uniformly oriented MTs.

The following paragraphs summarize the main findings in the manuscript, point out issues with the current version, and suggest approaches the authors could consider in terms of aligning the data and modeling to improve the manuscript.The key experimental findings and issues are:1. +end out microtubules are less likely to undergo catastrophe near the advancing axon tip, leading to unbounded MT growth.The critical issue here is that the modeling, as it stands, suggests unbounded growth is not very important for establishing a uniform microtubule array. Thus, while this is an interesting result, its importance for establishing MT orientation is unclear.

We strongly disagree with the reviewer, please see our comments above. Without unbounded growth, our simulations yielded a graded distribution of MTs along the axon with minus-end-out MTs in the proximal axon, see Figure S8A. Our modelling and experimental data together demonstrate that unbounded growth of +end out MTs near the axon tip is important for achieving a uniform plus-end-out out orientation of MTs along the entire length of the axon, irrespective of the magnitude of the effect of its isolated perturbation shown in Figure S8C. We extended the discussion to clarify this very important point (l.343ff).

2. Decreasing MT growth with NOC or osm+ disrupts MT orientationWhile this could be interpreted to mean that modulating MT dynamics alters MT orientation, because nocodazole is known to disrupt dynein, the effect of nocodazole on MT orientation could be occurring through a change in dynein activity rather than a change in MT assembly.

As mentioned above, we are not aware of any direct effect of Nocodazole on dynein-based MT sliding. In the literature, Nocodazole appears to disrupt dynein function mainly through dynein mislocalisation because of Nocodazole-mediated MT depolymerisation (Gerlitz et al., 2013). In our experiments, however, microtubules still polymerised (see Figure 3), so that it is unlikely that most of the dynein present becomes mislocalised. Furthermore, we are not aware of any direct effect of osmotic pressure on dynein activity. As we show a direct effect of both treatments on MT dynamics and orientation, and our modelling and simulation data strongly indicate that MT dynamics is important for achieving uniform MT orientation, our interpretation of the data seems much more plausible.

3. Disrupting p150 function reduces MT growth (dg um/cycle) in the last 10 μm of the axon and reduces MT orientation.As noted in the previous review, p150 influences dynein activity. Thus, the effects on MT orientation could be occurring through a change in dynein activity rather than MT dynamics.

As stated above and in our manuscript, to the best of our knowledge, it remains unclear whether p150 is required for MT-MT sliding. Furthermore, we provide direct evidence that perturbations of p150 significantly impact MT polymerisation dynamics, which our modelling and simulation data show to be important for setting up the uniform MT orientation in axons.

4. Disruption of dynein disrupts MT orientation, decreases MT growth (i.e., dg) in the last ten microns of the axon, but increases dg along the axon.The observation that disruption of dynein alters MT dynamics is interesting. Still, the observation that dg decreases near the growth cone and rises along the axon makes it challenging to interpret the results.

We agree with the reviewer that it is not straight-forward to interpret the rise of MT growth away from the axon tip. As dynein is required for the localisation of p150 to axon tips (Lazarus et al., 2013), dynein removal could lead to a more even distribution of p150 along the axon. The accompanying increase of p150 concentration away from the axon tip could potentially explain the observed increase of MT growth along the axon shaft.

5. Disruption of kinesin (khc-RNAi) reduces p150 levels in the growth cone, disrupts MT orientation, decreases MT growth (i.e., dg) in the last ten microns of the axon, but increases MT growth along the axon.Disruption of khc is known to disrupt dynein function in *Drosophila* neurons (Pilling et al., 2006, paper out of Saxton's lab). While the effects may be occurring by reducing p150 levels in the growth cone, a more straightforward explanation is that the disruption in MT orientation is occurring because disruption of kinesin disrupts dynein. Additionally, the decrease in dg near the growth cone and increase in dg along the axon is hard to interpret (Sup Figure 7F).

As the reviewer pointed out, kinesin is thought to transport dynein to MT +ends. Hence, it cannot be ruled out that kinesin knockdown affects microtubule sliding, which we already acknowledged in l.322ff. However, we fail to see why this should be the only effect of kinesin knockdown. Our data strongly suggest that also its effect on p150 accumulation in the growth cone, which is required for unbounded growth of plus-end-out MTs in that region, is significantly contributing to the perturbation of overall MT orientation.

6. Based on these observations and previous studies, the manuscript proposes a model for how axons achieve a uniform MT orientation. The modeling suggests that unbounded MT assembly works with MT sliding and templating to establish a uniform MT array.There are multiple issues with the modeling. The first is that the extent of rapid Stop and Go MT sliding in the experimental data set is not analyzed. From studies in *Xenopus* neurons from Popov's group (Ma et al., 2004 Cur Bio) and fly neurons (Gelfand's group) rapid sliding of MTs is exceedingly rare after neurite initiation. Based on this data, it seems likely that if the velocity distribution of comet motion were examined, essentially none of the comets would be moving at the rate of dynein/kinesin motors (i.e., ~ 1 um/sec). In turn, it follows a careful analysis of the Jupiter mcherry data might reveal that only tiny fraction, if any, of the MTs move via rapid MT sliding. I would suggest carefully looking at the experimental data and updating the model so it accurately reflects the observed extent of MT sliding.

This is an excellent point, which is in line with our data. We did not suggest that all MTs in the simulation undergo rapid transport, which, as the reviewer pointed out, has been shown to be exceedingly rare. In fact, we found no evidence of rapid movements in either the EB1-GFP data or the Jupiter mcherry data. In our simulations, most MTs moved with the growth velocity of the axon and showed zero relative velocity between themselves (as has been discussed in (Jakobs et al., 2015), see Figure 5B). This is consistent with a recent study by the Suter and Miller groups (Athamneh et al., 2017), which demonstrated that axonal MTs move as bulk with the growing velocity of the axon.

Sup Figure 8C suggests that unbounded MT growth alone does not contribute to microtubule orientation. Comparison of 8A, 8C, and 8E suggests unbounded growth modestly increases MT orientation in the proximal axon when sliding is included in the model. If one takes the model's output at face value, the verbal arguments in the manuscript claiming unbounded growth is essential for establishing MT orientation are undermined. I would suggest either accepting that the model indicates unbounded growth is relatively unimportant for establishing MT orientation or rethinking the model.

Please see our discussion above.

It's unclear why augmin is given such a prominent role in the model and the discussion. I found this distracting.

Our simulations demonstrated that augmin- mediated MT templating is essential for +end out MT orientation. It is as much required to achieve a uniform MT orientation in axons as dynein-based MT sliding and unbounded growth of plus-end-out MTs identified in the current study. Hence, we gave augmin a prominent role in the discussion as well.

In conclusion, there are two significant issues. The first is that manipulations designed to alter MT dynamics, either directly or by reducing the localization of p150 to the growth cone (i.e., p150, noc, kinesin), also disrupt dynein, which is well established in being essential for establishing MT orientation. As a result, the experimental data do not appear to show a definitive role for MT dynamics separate from dynein activity. The second is that the modeling suggests MT dynamics make a relatively minor contribution towards establishing MT orientation. In light of this, approaches that could be used to strengthen the manuscript would be to conduct experiments that more directly demonstrate the importance of MT dynamics in this process and to revise the model as outlined above.

We hope that our answers to the reviewer’s concerns above fully address these concerns. We have extended our discussion to avoid misunderstandings and confusion. All our experimental perturbations led to a change in MT dynamics and orientation, and our model and simulations show an important effect of MT dynamics on overall MT orientation in axons, leading to the first coherent model of how uniform MT orientation in axons arises.

Reviewer #3 (Recommendations for the authors):The authors have made additional efforts to address the comments raised by the reviewers, which has led to various improvements of their manuscript.

We would like to thank the reviewer for this positive statement.

In Supplemental Figure 5, the authors now report the fitted values and found a value for \α of 3.62. Given that d_g(x)=A*p150(x)^\α, this indicates that the gradient of microtubule behavior has a 4x steeper spatial decay than the gradient of p150. This could be the case, but it is not apparent from the stainings and profile plots of p150 they show in Figure 4C and S7. Based on Supplementary Figure 2, it seems that the p150 gradient is actually steeper.

We agree with the reviewer, the fitted data indicate that the spatial decay of dg is approximately 4x steeper than the p150 fluorescence profile. The exponent was found by fitting the equation in l.548 to the experimentally observed dg and p150 profiles from Figures 4C and S7. Figure 4—figure supplement 1 and Figure 4—figure supplement 7 show example profiles, with S2 exhibiting (as the reviewer pointed out) a steeper gradient than S9. If averaging over many profiles one arrives at Figure 4C

Furthermore, I had hoped that the authors would have added some reflection on how a gradient of p150 can result in a gradient of MT dynamics with a 4-fold different length scale. To me this is not entirely evident, but it could be due to a non-linear response of MT dynamics to p150 concentration.

We apologise for not addressing this apparent mismatch before. This non-linear dependence could result from the requirement of several p150 proteins to form a complex to affect MT catastrophe rates, as previously hypothesised (Lazarus et al., 2013). We added this interpretation to the discussion in line 308ff.

Unfortunately, the authors didn't try to address the dynamics of p150 to assess to which extent was diffuse versus anchored (e.g. using FRAP), and therefore the exact interplay between p150 and MTs at the axon tip remains unclear.

We agree that FRAP experiments to elucidate the diffusive or non-diffusive nature of p150 at axon tips would be interesting. We tried to address this point by expressing endogenously tagged p150 in our primary neuron culture system. Unfortunately, we were unable to obtain sufficient fluorescence intensity to do FRAP experiments. Since the question whether p150 is diffusive is not strictly relevant for our present manuscript, we decided to address this question in a future study.

Nonetheless, the manuscript does provide a more coarser insight into how uniform MT arrays could be established by local modulation of MT dynamics.

We thank the reviewer for appreciating a role of MT dynamics in setting up the uniform MT polarity found in axons.

References

Akhmanova, A., and Steinmetz, M.O. (2019). Microtubule minus-end regulation at a glance. J Cell Sci 132.

Athamneh, A.I.M., He, Y., Lamoureux, P., Fix, L., Suter, D.M., and Miller, K.E. (2017). Neurite elongation is highly correlated with bulk forward translocation of microtubules. Scientific Reports 7, 1–13.

del Castillo, U., Winding, M., Lu, W., Gelfand, V.I., and Allan, V. (2015). Interplay between kinesin-1 and cortical dynein during axonal outgrowth and microtubule organization in *Drosophila* neurons. *eLife* Sciences 4, e10140.

Gerlitz, G., Reiner, O., and Bustin, M. (2013). Microtubule dynamics alter the interphase nucleus. Cell. Mol. Life Sci. 70, 1255–1268.

Jakobs, M., Franze, K., and Zemel, A. (2015). Force Generation by Molecular-Motor-Powered Microtubule Bundles; Implications for Neuronal Polarization and Growth. Front Cell Neurosci 9, 441.

Lazarus, J.E., Moughamian, A.J., Tokito, M.K., and Holzbaur, E.L.F. (2013). Dynactin subunit p150(Glued) is a neuron-specific anti-catastrophe factor. PLoS Biol. 11, e1001611.

Ma, Y., Shakiryanova, D., Vardya, I., and Popov, S.V. (2004). Quantitative Analysis of Microtubule Transport in Growing Nerve Processes. Current Biology 14, 725–730.

Morrison, E.E., Moncur, P.M., and Askham, J.M. (2002). EB1 identifies sites of microtubule polymerisation during neurite development. Brain Res Mol Brain Res 98, 145–152.

Rao, A.N., Patil, A., Black, M.M., Craig, E.M., Myers, K.A., Yeung, H.T., and Baas, P.W. (2017). Cytoplasmic Dynein Transports Axonal Microtubules in a Polarity-Sorting Manner. Cell Rep 19, 2210–2219.

Tan, R., Foster, P.J., Needleman, D.J., and McKenney, R.J. (2018). Cooperative Accumulation of Dynein-Dynactin at Microtubule Minus-Ends Drives Microtubule Network Reorganization. Dev. Cell 44, 233–247.e234.

Waterman-Storer, C.M., Karki, S.B., Kuznetsov, S.A., Tabb, J.S., Weiss, D.G., Langford, G.M., and Holzbaur, E.L. (1997). The interaction between cytoplasmic dynein and dynactin is required for fast axonal transport. Pnas 94, 12180–12185.